# Visual data-type understanding does not emerge from scaling vision-language models

**Vishaal Udandarao**[1,2,*]   **Max F. Burg**[2,3,*]   **Samuel Albanie**[1,§]   **Matthias Bethge**[2,§]

[1] University of Cambridge, UK [2] University of Tübingen, Tübingen AI Center, Germany
[3] Institute of Computer Science and Campus Institute Data Science, University of Göttingen, Germany
[*] Equal contribution. Author ordering decided by coin flip. [§] Joint senior authors.
Correspondence: vu214@cam.ac.uk and max.burg@bethgelab.org

## ABSTRACT

Recent advances in the development of vision-language models (VLMs) are yielding remarkable success in recognizing visual semantic content, including impressive instances of compositional image understanding. Here, we introduce the novel task of *Visual Data-Type Identification*, a basic perceptual skill with implications for data curation (e.g., noisy data-removal from large datasets, domain-specific retrieval) and autonomous vision (e.g., distinguishing changing weather conditions from camera lens staining). We develop two datasets consisting of animal images altered across a diverse set of 27 visual *data-types*, spanning four broad categories. An extensive zero-shot evaluation of 39 VLMs, ranging from 100M to 80B parameters, shows a nuanced performance landscape. While VLMs are reasonably good at identifying certain stylistic *data-types*, such as cartoons and sketches, they struggle with simpler *data-types* arising from basic manipulations like image rotations or additive noise. Our findings reveal that (i) model scaling alone yields marginal gains for contrastively-trained models like CLIP, and (ii) there is a pronounced drop in performance for the largest auto-regressively trained VLMs like OpenFlamingo. This finding points to a blind spot in current frontier VLMs: they excel in recognizing semantic content but fail to acquire an understanding of visual *data-types* through scaling. By analyzing the pre-training distributions of these models and incorporating *data-type* information into the captions during fine-tuning, we achieve a significant enhancement in performance. By exploring this previously uncharted task, we aim to set the stage for further advancing VLMs to equip them with visual data-type understanding.

## 1 INTRODUCTION

Vision-Language Foundation Models (VLMs) (Bommasani et al., 2021) lie at the frontier of the machine learning ecosystem. Profiting from high-capacity transformer architectures (Vaswani et al., 2017) and large-scale pre-training, these models excel at identifying the semantic content in images (Radford et al., 2021; Pham et al., 2023; Jia et al., 2021). They also exhibit strong robustness to image distortions and perceptual changes as assessed on benchmarks like ImageNet-C (Hendrycks & Dietterich, 2019), ImageNet-Sketch (Wang et al., 2019), and ObjectNet (Barbu et al., 2019). Taking ImageNet-C as a concrete example, a classifier is tasked with correctly identifying a category (e.g., a *stingray*) in the presence of a particular data transformation (e.g., *defocus blur*). Similarly, the other domains and perceptual transformations contained in ImageNet-C, ImageNet-Sketch, and ObjectNet can be seen as examples of different *Visual Data-Types* obtained from ImageNet through applying image transformations that affect the appearance but not the content of the image.

The prevalent strategy in computer vision to cope with variable data-types is to use domain invariant classifiers, often achieved via data augmentation during training. An alternative strategy would be to retain the data-type specific information and explicitly model its composition with the semantic content of the image (Fig. 1A). This constitutes a symmetric split of the total image information into the complementary components of *semantics* and *visual data-types* (Granlund & Knutsson, 1995). Humans can flexibly switch between these two complementary aspects and visual data-type identification is an integral part of human perception (Oliva & Torralba, 2007; Ren et al., 2020; Bracci et al., 2023).

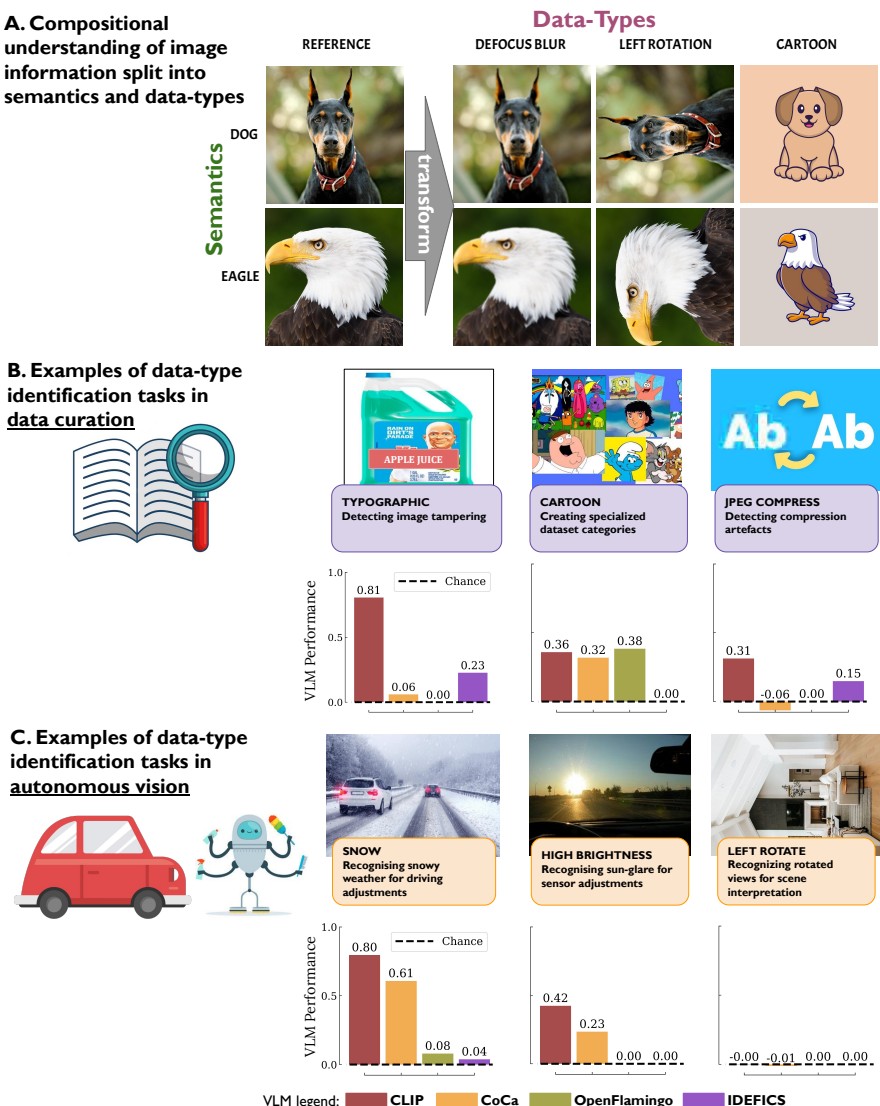

Figure 1: **Data-type identification highly impacts vision tasks.** Complementary to standard *semantic* recognition tasks (**A**), *data-type identification* targets recognising style and other contextual domain information. It is applicable for many practical scenarios, e.g., (**B**) data curation, and (**C**) autonomous cars and agents. In all contexts, flexible recognition of data-types is paramount, yet, VLMs exhibit poor performance on different data-types as illustrated by 4 select VLMs on 6 data-types (highlighted in the boxes). Notably, there is no one-size-fits-all model, underscoring the challenge of universal *data-type identification*. The bar plots report the informedness metrics, for more details refer to Sec. 4.1.

The recent breakthrough of large language models (LLMs) to mimic human text understanding is reflected by remarkable compositional reasoning skills and flexibility to cope with arbitrary contexts and textual data-types. This suggests that VLMs could also gain an increasingly flexible, compositional image understanding to cope with arbitrary visual data-types by inheriting it from the use of such powerful LLMs. Therefore, we seek to investigate to what extent the increasing robustness of VLMs to distribution shifts could be a consequence of compositional *data-type understanding*.

The most likely alternative would be that the increasing robustness of VLMs originates from increasing domain invariance (Mintun et al., 2021). However, VLMs differ in two important ways from ImageNet-trained classification models of the last decade: (1) They are trained on much more data crawled from the internet making it difficult to judge whether a test image is in-domain or

out-of-domain (Mintun et al., 2021; Fang et al., 2022; Nguyen et al., 2022), and (2) Due to the compositional nature of language, training on image-text-pairs could facilitate a compositional understanding of images in VLMs. Both points drive performance on a large set of visual benchmarks, yet, it is not easy to dissect their specific contributions. In addition, compositional understanding itself is a property that needs to be learned and thus expected to gradually improve with the amount of training data and model scale (Wiedemer et al., 2023).

Here, we test the hypothesis that dataset robustness of VLMs could be a consequence of compositional *data-type understanding* by creating a carefully designed *data-type identification* task and investigating to what extent VLMs exhibit a compositional understanding of semantic context and image appearance. Data-type identification is a necessary condition for data-type understanding: If a VLM understands the data-type of an image, e.g., the blurring operation, it needs to be able to identify it, independently from the particular image content.

Further, identifying the visual data-type of an image in addition to its semantic context is relevant in many real-world scenarios. For *(1) data curation and filtering* this is useful, for instance to exclude images of unwanted appearance from an image corpus (e.g., blurred samples), or to create a specific stylized domain generalization dataset (e.g., cartoons, sketches) (see Fig. 1B). In the context of *(2) autonomous vision* (e.g., self-driving cars, household robots), knowing the data-type of camera-scenes is relevant to interpret the data and intervene accordingly: for example, adapting driving style or sensor sensitivity based on detecting changing weather conditions versus sun-glare (see Fig. 1C).

Rather than engineering narrow solutions for each of these problems individually, the flexibility of VLMs affords a general ability to cope with all possible conditions. A compositional understanding of data-types would be an attractive solution to achieve this level of generality, and it could be highly useful for practical challenges such as the long-tailed test-time distribution encountered in autonomous driving (Dosovitskiy et al., 2017; Makansi et al., 2021; Zhou et al., 2022). Due to the combinatorial number of possible conditions and the open-ended nature of perception for an autonomous agent, the importance of a compositional understanding of data-types extends to robotics at large to deal with variable conditions in households, agriculture, or healthcare.

In summary, our work aims to make progress on *Data-Type Identification*; for this, we created two novel datasets containing images of animals, spanning 27 different data-types (see Fig. 2). On this data, we zero-shot benchmarked 39 state-of-the-art VLMs, with model sizes ranging from 100M to 80B parameters, across contrastive and LLM-based VLMs. We find that scaling up model size does not yield significant improvements. In particular, the largest auto-regressively trained VLMs perform significantly worse than their smaller contrastively-trained counterparts like CLIP. By investigating their performance across individual data-types, we found connections to structures in the pre-training data and vision embedding spaces of VLMs. Using this, we show that performance on the novel data-type identification task can be enhanced by fine-tuning with carefully tailored data. Our findings highlight an important limitation in the training of current leading VLMs: while they clearly excel on recognizing semantic content, acquiring data-type identification skills does not emerge from simply scaling up but rather requires a systematic change of the training data.

## 2 RELATED WORK

**Stress-testing VLMs.** Initial reports on the abilities of VLMs (e.g., in visual question answering) were largely anecdotal. Very recently, there is a growing interest in systematic investigation of such capabilities, often entailing the creation of synthetic datasets tailored for specific evaluations (Yuksekgonul et al., 2022; Parcalabescu et al., 2021; Thrush et al., 2022; Hsieh et al., 2023; Zhao et al., 2022; Lewis et al., 2022; Yamada et al., 2022; Ma et al., 2023; Kamath et al., 2023; Marathe et al., 2023; Yarom et al., 2023; Bitton-Guetta et al., 2023; Bordes et al., 2023). Here, we too synthesize a controlled dataset, but distinctively introduce the new task of *Data-Type Identification*, a basic perceptual skill that remains largely underexplored in previous work.

**Distribution shifts, anomaly detection and domain adaptation.** While many existing approaches study perceptually altered data, e.g., distribution shifts (Hendrycks et al., 2021; Taori et al., 2020; Schneider et al., 2020; Qiu et al., 2022; Rauber et al., 2017; Koh et al., 2021), domain adaptation (Farahani et al., 2021; You et al., 2019; Wang & Deng, 2018), out-of-distribution detection (Hendrycks & Gimpel, 2016; Yang et al., 2021; Liu et al., 2020), and anomaly detection (Roth

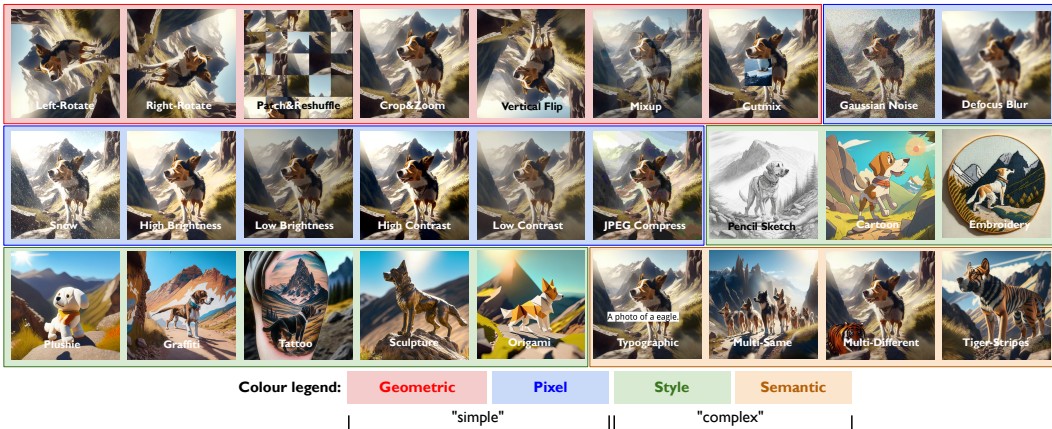

Figure 2: **Proposed data-types.** Example images from our *SyntheticTypeIdent* dataset for each of our 27 data-types, spanning four categories: geometric, pixel, style, and semantic data-types.

et al., 2022; Han et al., 2022; Pang et al., 2021), they often only determine the presence of an anomaly or shift without pinpointing its exact nature. In contrast, if an intervention to an anomaly is necessary, we need to pinpoint its exact nature, which is the goal of *Data-Type Identification*.

Very few previous works have touched upon this question in narrow scenarios. Some studied identifying few specific perceptual data-types using convolutional neural networks (CNNs) in a binary classification setup, e.g., image mirroring (Lin et al., 2020) and cropping (Van Hoorick & Vondrick, 2021). Zhu et al. (2022) trained linear probes to understand predictability of domains like paintings or cartoons from the image features of a pre-trained CNN. Paiss et al. (2023) investigated counting objects in VLMs (similar to our MULTI_SAME and MULTI_DIFFERENT data-types, see Fig. 2). An et al. (2023) showed that CLIP can reasonably infer a limited number of simple data-types in a binary classification setup and used this to improve CLIP's zero-shot semantic classification. Rashtchian et al. (2023) used linear probes on the image embedding spaces of vision-only and vision-language models, to identify perceptual manipulations on images, without accounting for their text encoders. Our *Data-Type Identification* framework subsumes all these setups in a unifying approach: we investigate an extensive range of 27 *data-types* across a broad perceptual and stylistic range for 39 VLMs, encompassing both contrastively-trained discriminative models and auto-regressively trained generative models. Our work therefore enables studying generalisation of VLMs on a broad set of data-types.

## 3  THE TYPEIDENT DATASETS

To probe the effectiveness of VLMs in identifying data-types, we created two novel datasets consisting of images of a single animal in a scene, spanning 27 data-types across 4 categories: **geometric** (e.g., left-rotation), **pixel** (e.g., applying Gaussian noise), **style** (e.g., creating a cartoon-version), and **semantic** (e.g., replacing a single animal with multiple animals). Note that geometric and pixel data-types can be obtained from simple, well-defined transformations such as pixel re-arrangement, linear filtering, or additive noise. In contrast, most transformations generating different style and semantic data-types from a reference image distinctively rely on the use of more complex neural networks. For a complete list of all data-types studied, see Fig. 2 and refer to the Appendix.

Our first dataset, *SyntheticTypeIdent*, is constructed by first generating a set of 50 reference-images of animals using a text-to-image model, with these images uniformly sampled across 10 animal species. Each generated reference-image was then altered with all our 27 data-type transformation functions, resulting in 1,350 evaluation samples (see Fig. 2 for an example of all data-type transformations). For creating the geometric and pixel data-type images, we directly applied the corresponding point-wise image transformation function (e.g., adding Gaussian noise) on the reference-images. To precisely control the transformations applied, we regenerated style and most semantic-level data-type images again using the same diffusion model.

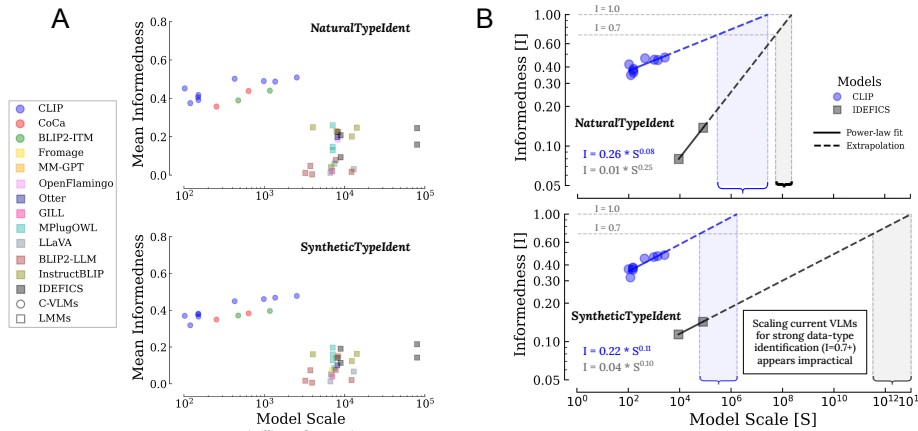

Figure 3: **(A) VLMs struggle with identifying data-types.** Less recent, contrastively learned C-VLMs (e.g., CLIP) outperform the much larger and more recent LMMs (e.g., IDEFICS) despite the latter's strong language model priors. Scaling shows limited effect on performance. Chance-level performance is at 0. **(B) Weak scaling laws for VLMs.** Power-law fits reveal that for achieving strong data-type identification (mean informedness>0.7), current VLMs would need to surpass a trillion parameters. This calls for an alternative strategy to just scaling up current VLMs.

For our second dataset, *NaturalTypeIdent*, we manually curated 50 reference-images from KaggleAnimalImages (Banerjee, 2023). We then followed the exact same procedure for creating data-type images from the reference-images. However, all generative steps were replaced by a refined, deduplicated web-retrieval step for mining style and semantic data-type images. This provides an in-the-wild, naturally occurring testbed, thereby complementing the precisely controlled *SyntheticTypeIdent* dataset. Since we can procure appropriate images for only 25 data-types (we omit MULTI_DIFFERENT and TIGER_STRIPES), *NaturalTypeIdent* only contains 1,250 samples.

Importantly, we manually verified both datasets to ensure that the target data-type for each image was the most prominent data-type reflected in it, enabling a careful study between models without interference between data-types. For details about dataset creation refer to the Appendix.

## 4 BENCHMARKING VLMs ON DATA-TYPE IDENTIFICATION

### 4.1 EXPERIMENTAL SETUP

We evaluated 39 VLMs from 13 model families, with sizes ranging from 100M to 80B parameters, across two groups: discriminative, contrastively-trained VLMs (e.g., CLIP) which we refer to as **C-VLMs**, and generative, auto-regressively trained VLMs (e.g., OpenFlamingo) which we refer to as large multi-modal models (**LMMs**) (Li, 2023). Specifically, from the C-VLM group we evaluated CLIP (Radford et al., 2021), BLIP-2-ITM (Li et al., 2023c), and CoCa (Yu et al., 2022); in the LMM group we tested Fromage (Koh et al., 2023b), GILL (Koh et al., 2023a), Multimodal-GPT (Gong et al., 2023), OpenFlamingo (Awadalla et al., 2023), Otter (Li et al., 2023a), MPlugOwl (Ye et al., 2023), LLaVA (Liu et al., 2023a), BLIP-2-LLM (Li et al., 2023c), InstructBLIP (Dai et al., 2023), and IDEFICS (Laurençon et al., 2023). We tested all VLMs on correctly classifying the target data-type for each evaluation image, in a zero-shot manner. We evaluated C-VLMs by computing the cosine-similarity of the image embedding and the text embedding of the specific data-type description, e.g., "A blurred image of an animal." (see Appendix for full list). For a fair comparison, we evaluated LMMs by log-likelihood scoring (Dai et al., 2023; Li et al., 2023b) each of the 27 data-type description texts, with the prompt: "<image> Q: Describe the image. A: <data_type_description>", replacing <data_type_description> by the corresponding text description for a particular data-type. We quantified model performance using informedness, $I_k=\text{TPR}_k-\text{FPR}_k$ on data-type $k$, which in addition to the true positive rate (TPR, i.e., accuracy) accounts for the false positive rate (FPR). We summarized model performance as mean informedness across data-types, $\mu_I=\langle I_k\rangle_k$. See Appendix for evaluation details.

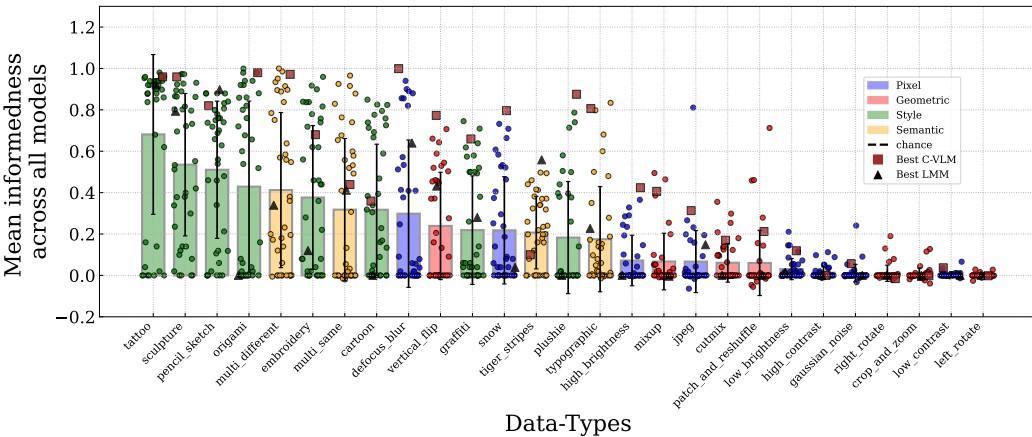

Figure 4: **Average-performance across data-types on *SyntheticTypeIdent*.** VLMs perform reasonably on style and semantic data-types (e.g., PENCIL_SKETCH, CARTOON) and show weak results on pixel and geometric data-types (e.g., GAUSSIAN_NOISE, HIGH_CONTRAST). Chance-level at 0.

## 4.2 VLMs STRUGGLE WITH IDENTIFYING DATA-TYPES

Our evaluations reveal that all tested VLMs exhibit limited performance on both *SyntheticTypeIdent* and *NaturalTypeIdent* (Fig. 3A). We found that C-VLMs performed better than LMMs, even though the latter are more recent and orders of magnitude larger. The best C-VLM achieved mean informedness $\mu_I = (0.47, 0.50)$ while its LMM counterpart achieved $\mu_I = (0.22, 0.25)$ on *SyntheticTypeIdent* and *NaturalTypeIdent*, respectively. As a control and for direct comparison, we also tested models on animal identification on *SyntheticTypeIdent*. As expected, the performance on this semantic recognition task is very good, achieving a mean informedness across models of 0.89. This confirms quantitatively that the performance on identifying data-types (detailed plots in Appendix) is substantially worse than on object recognition. We further note three key findings from our evaluations:

**LMMs, a downgrade?** Surprisingly, LMMs consistently underperform C-VLMs, despite using LLMs as text models, compared to the smaller text encoders in C-VLMs. Notably, the largest LMM (IDEFICS, 80B parameters) substantially underperforms an orders-of-magnitude smaller CLIP-RN50 (100M parameters). The rich language grounding that LLMs inherit from extensive real-world text training seemingly does not provide benefits for identifying data-types. This result challenges the prevailing notion that strong language model priors can improve fine-grained understanding in VLMs (Cascante-Bonilla et al., 2023; Doveh et al., 2023; Yuksekgonul et al., 2022; Wang et al., 2023). We hypothesise two plausible causes for this performance drop to be studied in detail by future work: (1) *Weak alignment* between the vision encoder and LLM might degrade the real-world symbolic grounding innate to each independently (Bavishi et al., 2023). (2) *Discriminative-Generative gap* might be at play, i.e., discriminating between answers is easier than generating one (Vapnik, 1999; Ng & Jordan, 2001). Both suggest that C-VLM contrastive objectives might better equip them for data-type identification than LMM auto-regressive objectives (Liu et al., 2023b).

**Weak scaling behaviour.** Interestingly, within the C-VLM and LMM groups, our results suggest weak scaling effects. We analysed this quantitatively by fitting a power-law (Alabdulmohsin et al., 2022; Henighan et al., 2020; Cherti et al., 2023) on the observed mean informedness vs. model scale relationship for CLIP (C-VLM) and IDEFICS (LMM), since they span the widest parameter sizes within a model family. Fig. 3B confirms the weak scaling law, indicating a severe limitation for current VLMs: to achieve a performance practicable for data-type identification ($\mu_I > 0.7$), current models would need to surpass a trillion parameters. This calls into question the effects of model scaling, and whether alternate strategies are required to enhance their performance.

**Stark performance differences between simple and complex data-types.** To get a finer-grained understanding of the overall model performance (Fig. 4) we break-down the per-data-type averaged mean informedness across all models. We find that while VLMs are reasonably good at identifying style and semantic data-types, they falter systematically on pixel and geometric data-types. For the majority of data-types even the best-performing models struggle to surpass chance-level perfor-

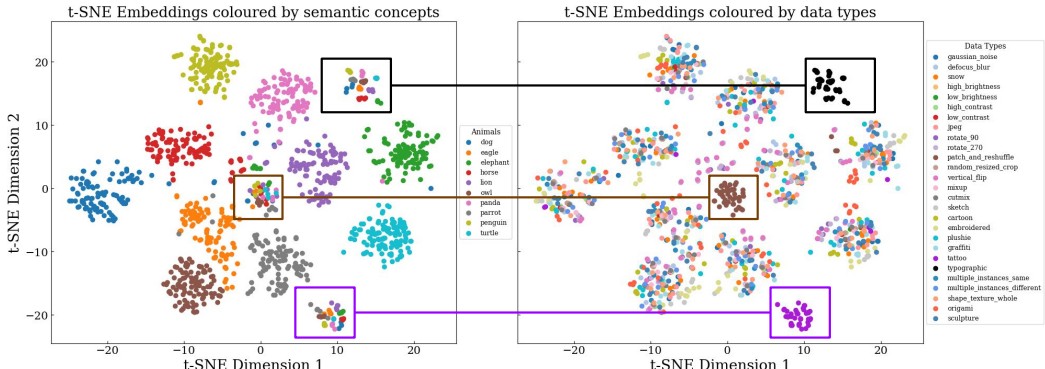

Figure 5: **What does CLIP's image embedding space encode?** CLIP-RN50's image embeddings, colour-coded by ground-truth semantic concept (left) and data-type (right), reveal its pronounced affinity for recognising semantic concepts, while being largely invariant to data-type distinctions.

mance and no single model consistently outperforms others across a majority of data-types. Instead, multiple models each excel in identifying just a few specific data-types. This reveals inherent biases in the pre-training procedures of VLMs, limiting the desired generality of foundation models.

## 5 UNDERSTANDING WHY VLMs UNDERPERFORM IN IDENTIFYING DATA-TYPES

We next investigate two plausible reasons for the sub-par performance of VLMs in identifying data-types: (1) their image embeddings lack data-type discriminative information, and (2) their pre-training datasets, despite the enormous sizes, lack sufficient data-type specific information, limiting models from learning data-type discriminative features. We probe both candidate reasons in detail, performing a case study with CLIP, and find good evidence for both of them. Due to CLIP being a prototypical C-VLM, and the widespread adoption of its vision encoders in LMMs, we suggest that our findings should be broadly applicable.

**Reason 1: Peeking into CLIP's embedding space.** We visualized the CLIP image embeddings of *SyntheticTypeIdent* using t-SNE (Van der Maaten & Hinton, 2008). Colour-coding the embeddings by (1) the image's semantic concept, i.e., the animal type (Fig. 5 left), and (2) the image's target data-type (Fig. 5 right), uncovered an interesting dichotomy: while distinct embedding clusters emerge based on semantic concepts (animals), most data-types are not clearly demarcated (see Appendix for KNN and linear-probe analysis). This suggests that CLIP's vision encoder is somewhat invariant to data-types, despite it not being explicitly trained to be so (only random-resized cropping was used as training data-augmentation, discussion in Appendix). As most C-VLMs and LMMs use CLIP image embeddings, this potentially explains the poor performance of all VLMs on identifying data-types. We further note that the embeddings of only three data-types are closely clustered (TATTOO, PATCH_AND_RESHUFFLE, and TYPOGRAPHIC), yet, these are precisely the embeddings which are not directly semantically distinguishable—this suggests that CLIP might not encode semantic and data-type information compositionally but rather sacrifices one (data-type) over the other (semantics). This offers a consistent explanation why CLIP models are so effectively robust at classifying semantic content (Fang et al., 2022; Shi et al., 2023; Nguyen et al., 2022; Santurkar et al., 2022; Ramanujan et al., 2023) but fail at solving the complementary problem of data-type identification.

**Reason 2: Peeking into VLM pre-training datasets.** Fig. 4 revealed that VLMs fare well on some complex data-types while falling short on simple ones. An intuitive explanation is pre-training dataset imbalance: an abundance of samples aligning with style data-types (e.g., CARTOON, PEN-CIL_SKETCH) and a paucity of simple data-types (e.g., GAUSSIAN_NOISE, LEFT_ROTATE). To confirm this quantitatively, we analysed LAION-2B-en—CLIP's pre-training dataset. We first counted and retrieved all samples containing representative data-type keywords in the captions (e.g., "blurry"; see Appendix for details and a semantic search-based analysis). As pure keyword-frequency might not account for mis-aligned image-caption pairs, we estimated an *alignment prob-*

*ability*—the fraction of retrieved samples where the image aptly captures the data-type concept—by manually labeling 100 random samples per data-type for data-type accuracy. Finally, we computed an *abundancy score* as the product of text-frequency and *alignment probability*. Correlating this *abundancy score* with averaged model performance across data-types revealed strong positive associations (Spearman rank correlation, $r=0.557$ for *SyntheticTypeIdent*; $r=0.489$ for *NaturalTypeIdent*). The association is even stronger on *SyntheticTypeIdent* when correlating *abundancy score* with CLIP-model averaged performance ($r=0.606$), suggesting that the varying model performance across data-types can be explained by the constraints of their pre-training data distribution.

## 6 IMPROVING VLMS TO IDENTIFY DATA-TYPES

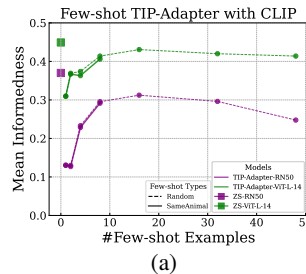

(a)

Having understood some factors limiting the performance of VLMs, we experiment with methods using data-type information-rich samples to improve them. Here, we investigate CLIP (C-VLM) and Otter (LMM) as two representative models.

### 6.1 FEW-SHOT TRAINING-FREE ADAPTATION DOES NOT HELP

Can few-shot examples boost performance without updating model weights, using in-context learning (Dong et al., 2022; Brown et al., 2020) or training-free adapters (Zhang et al., 2021; Udandarao et al., 2022)? We answer next.

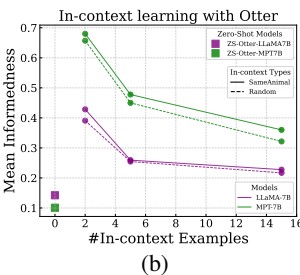

(b)

**CLIP TIP-Adapter.** We test the TIP-Adapter (Zhang et al., 2021) framework with CLIP, using two few-shot example selection strategies: *Random* (selecting examples with random animals) and *SameAnimal* (selecting examples with same animal as test image). We evaluate $1, 2, 4, 8, 16, 32, 48$ shots with RN50 and ViT-L-14 vision encoders. We found few-shot adaptation degrading performance across all settings (see Fig. 6a). This presumably originates from TIP-Adapter leveraging semantic similarities in CLIP's image embedding space, which lacks information to disambiguate between data-types (see Fig. 5). Hence, TIP-Adapter cannot capture any information discriminative across data-types but rather exploits semantic similarities b/w concepts which is detrimental for our task.

Figure 6: **Few-shot training-free adaptation methods fail.** Both TIP-Adapter with CLIP (top) and in-context learning with Otter (bottom) fail to substantially improve VLM data-type identification.

**Otter In-context Learning.** We explored various in-context example selection strategies and found selecting $n$ examples with one whose data-type matched the target of the test sample and other $n-1$ randomly worked the best—we evaluate $n=2,5,15$ examples on the *Random* and *SameAnimal* strategies, using LLaMA-7B (Touvron et al., 2023) or MPT-7B (MosaicML, 2023) as LLM-backbones (see Appendix for details and in-context scaling results with LLaVA). Surprisingly, we found an initial uptick in performance with $n=2$, followed by a decline as in-context examples increased (see Fig. 6b). We attribute this to Otter overfitting on its in-context examples, i.e., simply predicting a random data-type from within the in-context examples. Since chance-level performance also increases with fewer in-context examples, this could explain improved performance with $n=2$. We conclude that in-context learning does not enhance Otter's ability to identify data-types.

**Takeaways.** Our empirical results strongly indicate that training-free few-shot approaches fail to enhance VLMs for identifying data-types, likely because VLMs lack data-type discriminative information in their embeddings. Rather, an intensive training procedure to infuse data-type knowledge might be more promising.

### 6.2 FINE-TUNING WITH APPROPRIATE DATA-MIXTURES IMPROVES PERFORMANCE

**Data-mixtures.** We created a specialised dataset, *TeDaTy* (Teaching Data-Types), incorporating data-type information into images and text-captions. We construct training images, sourced from COCO (Lin et al., 2014), ImageNet (Deng et al., 2009), PACS (Li et al., 2017), and DomainNet (Peng et al., 2019), by applying our data-type transformation functions and adapting the captions

Table 1: **CLIP ViT-B-32 fine-tuning results on *TypeIdent* datasets with different data-mixtures.**

| Data-Mixture | *SyntheticTypeIdent* | | | | | | *NaturalTypeIdent* | | | | | |
|---|---|---|---|---|---|---|---|---|---|---|---|---|
| | **Full** | | **Freeze-Image** | | **Freeze-Text** | | **Full** | | **Freeze-Image** | | **Freeze-Text** | |
| | ID-I | OOD-I | ID-I | OOD-I | ID-I | OOD-I | ID-I | OOD-I | ID-I | OOD-I | ID-I | OOD-I |
| Zero-shot CLIP | 0.451 | 0.457 | 0.451 | 0.457 | 0.451 | 0.457 | 0.440 | 0.473 | 0.440 | 0.473 | 0.440 | 0.473 |
| COCO (control) | 0.451 | 0.468 | 0.354 | 0.465 | 0.488 | 0.451 | 0.494 | 0.507 | 0.451 | 0.500 | 0.457 | 0.473 |
| TeDaTy | 0.669 | 0.392 | 0.777 | 0.469 | 0.780 | 0.370 | 0.691 | 0.412 | 0.654 | 0.474 | 0.646 | 0.379 |
|    + COCO | 0.646 | 0.394 | 0.717 | 0.465 | 0.631 | 0.371 | 0.629 | 0.400 | 0.680 | 0.470 | 0.574 | 0.356 |
|    + COCO + IN100k | 0.600 | 0.383 | 0.700 | 0.469 | 0.586 | 0.354 | 0.557 | 0.381 | 0.634 | 0.456 | 0.471 | 0.323 |

Table 2: **Otter-LLaMA-7B fine-tuning results with different data-mixtures.**

| Data-Mixture | *SyntheticTypeIdent* | | *NaturalTypeIdent* | |
|---|---|---|---|---|
| | ID-I | OOD-I | ID-I | OOD-I |
| Zero-shot Otter | 0.051 | 0.180 | 0.102 | 0.256 |
| COCO (control) | 0.020 | 0.246 | 0.085 | 0.315 |
| TeDaTy | 0.088 | 0.061 | 0.111 | 0.111 |
|    + COCO | 0.106 | 0.168 | 0.171 | 0.276 |
|    + COCO + IN100k | 0.120 | 0.166 | 0.166 | 0.261 |

accordingly, e.g., "This is a cartoon image of a dog.". TeDaTy comprises 8 in-distribution (ID) data-types, holding out 19 for out-of-distribution (OOD) generalisation tests (see Appendix for details). To isolate effects of data-distributions, we experiment with three data-mixtures: (1) TeDaTy, (2) TeDaTy+COCO, and (3) TeDaTy+COCO+IN100k (sub-sampled from ImageNet). We also fine-tune only on COCO as a control to disentangle gains from fine-tuning and specific data-mixtures.

**Results.** Fine-tuning CLIP improved performance on the ID data-types for all TeDaTy mixtures (Tab. 1). However, COCO-only fine-tuning degraded ID-performance, highlighting the importance of incoporating key data-type information with TeDaTy. Freezing the vision-encoder while fine-tuning provided large ID-boosts and surprisingly even improved OOD. Freezing the text-encoder improved ID-performance but degraded OOD-performance, likely because of large gradients from only updating the vision-encoder. This corroborates previous CLIP-tuning studies (Zhai et al., 2022).

**Transfer to Otter.** To fine-tune Otter, we kept the vision encoder frozen (best CLIP fine-tuning strategy) and tuned only the perceiver resampler, cross-attention and embedding layers. We found fine-tuning with all TeDaTy variants improved ID-performance up to two-fold, while preserving OOD-performance (see Tab. 2). Fine-tuning only with COCO degrades ID-performance, reinforcing the importance of a dataset that captures data-type knowledge.

**Takeaways.** Our results suggest that training with data-mixtures explicitly inducing data-type information is a promising direction for improving VLM data-type identification.

# 7 CONCLUSION

In this work, we introduced and motivated *Data-Type Identification* as a basic perceptual skill with general implications for visual foundation modeling. We created two novel datasets to study model performance on this task, and released a third dataset tailored to fine-tune models to improve data-type identification. Our extensive zero-shot experiments across 39 VLMs revealed that they struggle to identify many data-types. Interestingly, scaling model size results only in minimal gains—we traced this back to the structure in VLM embedding spaces and pre-training datasets, and suggest that studying weak alignment between image-encoders and LLMs (Bavishi et al., 2023) as well as the discriminative-generative gap (Vapnik, 1999; Ng & Jordan, 2001; Saunders et al., 2022) will be promising directions for future work (see for example Liu et al. (2023b)). We found that training-free few-shot methods do not improve performance, and that it is necessary to incorporate data-type information back into the training process. Taken together, our study reveals an important limitation of the desired generality of foundation models, and the dataset and insights presented in this paper set the stage for further advancing VLMs for visual data-type understanding.

## REPRODUCIBILITY STATEMENT

We provide code and datasets to reproduce all experiments in the paper here: `https://github.com/bethgelab/DataTypeIdentification`. For the *TypeIdentDatasets*, we have provided comprehensive details on dataset creation in the Appendix. We specify the details of the 39 models tested along with their evaluation methods in the Appendix. For all our fine-tuning experiments, we used a fixed random seed for reproducibility. Further, we will release all our fine-tuned checkpoints and make public the WeightsAndBiases training logs for easy access.

## ACKNOWLEDGEMENTS

The authors would like to thank (in alphabetic order): Alexander S. Ecker, Çağatay Yıldız, Evgenia Rusak, Roland Zimmermann, Shyamgopal Karthik, Surabhi S. Nath, Susanne Keller and Thomas Klein, for helpful comments and feedback. The authors thank the International Max Planck Research School for Intelligent Systems (IMPRS-IS) for supporting VU and MFB. VU thanks the European Laboratory for Learning and Intelligent Systems (ELLIS) PhD program for support. SA is supported by a Newton Trust Grant. This work was supported by the German Research Foundation (DFG): SFB 1233, Robust Vision: Inference Principles and Neural Mechanisms, TP4, project number: 276693517. MB is a member of the Machine Learning Cluster of Excellence, funded by the Deutsche Forschungsgemeinschaft (DFG, German Research Foundation) under Germany's Excellence Strategy – EXC number 2064/1 – Project number 390727645.

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

## A  DETAILS ABOUT TYPEIDENT DATASETS

In this section, we provide additional details on our dataset generation process.

### A.1  LIST OF DATA-TYPES STUDIED

In the table below, we provide a complete list of all the data-types we used in our study across the four coarse-grained categories.

Table 3: **The 27 data-types used in our study.**

| Geometric | Pixel | Style | Semantic |
|---|---|---|---|
| LEFT_ROTATE | GAUSSIAN_NOISE | PENCIL_SKETCH | TYPOGRAPHIC |
| RIGHT_ROTATE | DEFOCUS_BLUR | CARTOON | MULTI_SAME |
| PATCH_AND_RESHUFFLE | SNOW | EMBROIDERY | MULTI_DIFFERENT |
| CROP_AND_ZOOM | HIGH_BRIGHTNESS | PLUSHIE | TIGER_STRIPES |
| VERTICAL_FLIP | LOW_BRIGHTNESS | GRAFFITI | |
| MIXUP | HIGH_CONTRAST | TATTOO | |
| CUTMIX | LOW_CONTRAST | SCULPTURE | |
| | JPEG_COMPRESS | ORIGAMI | |

### A.2  ANIMALS USED IN OUR DATASETS

Our justification for selecting animal classes as the semantic concepts for our dataset was two-fold: (1) they are easy semantic concepts for most VLMs to identify, and (2) we can get a diverse set of animals ranging from mammals to birds such that our dataset is not too restricted to a particular class. The 10 animal classes we used for creating our datasets are: DOG, EAGLE, ELEPHANT, HORSE, LION, OWL, PANDA, PARROT, PENGUIN and TURTLE.

### A.3 SYNTHETICTYPEIDENT CONSTRUCTION

For constructing the dataset, we first need a set of 50 reference-images. For each of the 10 animal classes, we prompted ChatGPT for a set of 5 different prompt descriptions to use as text-prompts for feeding into a text-to-image diffusion model. The prompt descriptions we used are:

1. DOG:
   - "A captivating, photorealistic 8k image of a (dog:1.3) relaxing in front of a cozy fireplace in a log cabin during a snowy evening."
   - "A stunningly detailed 8k image of a (dog:1.3) hiking through a mountain range on a sunny day."
   - "A stunning, photorealistic 8k image of a (dog:1.3) sleeping peacefully under a tree in a lush garden."
   - "A natural and realistic 8k image of a (dog:1.3) lying on a carpet in front of a roaring fireplace in a cozy cabin."
   - "A breathtaking, high-resolution 8k image of a (dog:1.3) peacefully sitting at a campsite in front of a roaring bonfire under a starry sky."

2. EAGLE:
   - "An awe-inspiring 8k image of an eagle perched on a rocky ledge overlooking a vast desert landscape."
   - "A stunning, high-resolution 8k image of an eagle soaring majestically over a snow-capped mountain range."
   - "An exquisite, high-quality 8k image of an eagle soaring in a clear blue sky over a mountain range."
   - "A captivating, photo-realistic 8k image of an eagle perched on a tree branch overlooking a stunning waterfall."
   - "An ultra-high-resolution 8k image of an eagle perched on a branch, watching attentively for prey in a dense forest."

3. ELEPHANT:
   - "A vivid image of an elephant carrying logs in the midst of a busy local village."
   - "An 8k image of an elephant enjoying a mud bath in the middle of the serene savannah."
   - "A stunningly detailed, high-resolution image of a (elephant:1.3) taking a refreshing dip in a cool, clear river."
   - "An ultra-high-resolution image of an elephant crossing a river with tall grass swaying in the background."
   - "A detailed, high-quality 8k image of a (elephant:1.3) trumpeting loudly, while its herd is crossing a river flowing through a beautiful and serene landscape."

4. HORSE:
   - "A vibrant 8k image of a (horse:1.3) galloping through a shallow lake with a mountain range in the background."
   - "An enchanting 8k image of a majestic horse standing under a blossoming cherry tree with a tranquil pond and distant mountains in the background."
   - "An ultra-realistic 8k image of a majestic black horse grazing gently in a lush green meadow surrounded by a dense forest."
   - "A high-quality, photorealistic 8k image of a (horse:1.3) galloping freely on a deserted beach at sunset."
   - "A beautiful, realistic 8k image of a (horse:1.3) grazing in a green meadow with mountains in the background."

5. LION:
   - "A majestic 8k image of a (lion:1.3) standing tall on a rocky outcrop overlooking a vast desert landscape."

- "An impressive, photorealistic 8k image of a (lion:1.3), gazing majestically into the camera against a backdrop of a pale blue sky."
- "A captivating, high-resolution 8k image of a (lion:1.3) resting on a rock in front of a scenic waterfall."
- "A stunning, photorealistic 8k image of a (lion:1.3) relaxing in the shade of a tree on a hot summer day.,"
- "A dramatic 8k image of a (lion:1.3) roaring fiercely with a backdrop of a stormy sky.,"

6. OWL:

- "A stunning, photo-realistic 8k image of an owl (species TBD) swooping down to catch its prey during the night in a forest.,"
- "An incredibly lifelike 8k image of a (great horned owl:1.3) perched on a tree branch in a dense fog.,"
- "A beautiful and naturalistic 8k image of a (screech owl:1.3) nesting in a tree hole in a forest.,"
- "A captivating 8k image of an owl staring intently into the camera in a dimly lit forest."
- "An intricately detailed 8k image of an owl perched on a barren, windswept cliff in a stormy sky."

7. PANDA

- "An ultra-high-resolution 8k image of a (panda:1.3) lazily lying on a tree branch, basking in the sun's rays."
- "An impressive, high-quality 8k image of a panda (with panda colors: 1.3) standing in front of a gorgeous waterfall in a tropical jungle."
- "An ultra-realistic, photorealistic 8k image of a panda (1.3) playing with a ball in a grassy meadow under a sunny sky."
- "An attention-grabbing, photorealistic 8k image of a panda (1.3) standing tall on its hind legs and reaching for a juicy bunch of bamboo shoots."
- "A picturesque 8k image of a panda (1.0) stargazing while sitting on a grassy hilltop in a rural valley."

8. PARROT:

- "An ultra-high-resolution 8k image of a (parrot:1.3) enjoying some refreshing mist on a hot day near a waterfall in a mountainous forest."
- "A realistic 8k image of a (parrot:1.3) perched on a tree branch, with a lush jungle background."
- "An enchanting and realistic 8k image of a parrot taking a refreshing bird bath in a small stream, surrounded by wildflowers."
- "A breathtaking 8k image of a (parrot:1.3) perched on a wire fence with a stunning mountain range in the background."
- "A realistic 8k image of a (parrot:1.3) playing with ropes and swings in a well-furnished living room."

9. PENGUIN

- "A detailed, high-quality 8k image of a penguin waddling along a rocky beach with waves crashing in the background."
- "A stunning, high-resolution image of a (penguin:1.3) waddling on a snowy beach, with snow-capped mountains in the background."
- "A captivating, photorealistic 8k image of a (penguin:1.3) glancing over its shoulder, watching for predators amidst the Antarctic wilderness."
- "A breathtaking 8k image of a (penguin:1.3) standing on a rocky outcrop, overlooking the vast ocean with icebergs in the backdrop."
- "An exquisite, high-quality 8k image of a (penguin:1.3) waddling on a snowy mountain top against a clear sky background."

10. TURTLE

- "A stunning, high-quality 8k image of a (turtle:1.3) basking in the sun on a sandy beach."

- "An impressive, photorealistic 8k image of a (turtle:1.3) climbing up a steep mountain."
- "A beautiful, photorealistic 8k image of a (turtle:1.3) crawling through lush green foliage in a tropical rainforest."
- "An impressive, photorealistic 8k image of a (turtle:1.3) crawling on a log in a murky swamp."
- "A stunning, high-quality 8k image of a (turtle:1.3) basking in the sun on a rocky beach."

We then used the Kandinsky-2.1 text-to-image diffusion model to generate 50 synthetic reference-images (100 diffusion steps, 4 guidance-scale (Ho & Salimans, 2022), 768×768 output resolution) using the text prompts specified above. We transformed these reference-images into our final data-type images by leveraging pointwise transformation functions (for pixel and geometric data-types) or re-generating them using Kandinsky2.1 with the same random-seed but with a slight modification of the prompt to capture the specific data-type information. This results in 1,350 total evaluation samples across 27 data-types. We describe the specific functions and the prompt modifications applied for transforming the reference-images to data-type images across the four coarse-grained data-type categories below:

A. **Geometric**

- LEFT_ROTATE: We rotate every reference-image to the left ($90°$).
- RIGHT_ROTATE: We rotate every reference-image to the right ($270°$).
- PATCH_AND_RESHUFFLE: We patchify the image into a $5 \times 5$ grid, and then randomly shuffle them spatially.
- CROP_AND_ZOOM: We use a zoom-factor of 2 for randomly zooming into a reference-image.
- VERTICAL_FLIP: We flip every reference-image vertically.
- MIXUP: We mix every reference-image with another random reference-image with a mixing coefficient of 0.35.
- CUTMIX: On every reference-image, we paste a randomly cropped patch from another random reference-image.

B. **Pixel**

- GAUSSIAN_NOISE: To every reference-image, we add gaussian-noise such that its pixel-variance is increased $1.4\times$.
- DEFOCUS_BLUR: We iteratively blur the reference-image with a disk-shaped filter until it reaches a target blur level of $1.4\times$ the estimated initial blurriness level (estimated using normalized maximum laplacian variance).
- SNOW: We apply a randomized snow effect to the reference-image, by first generating a snow layer, followed by motion blur, and then overlaying back on the original image.
- HIGH_BRIGHTNESS: We increase the brightness of each reference-image multiplicatively by $1.5\times$ in the HSV colour space.
- LOW_BRIGHTNESS: We reduce the brightness of each reference-image multiplicatively by $1.5\times$ in the HSV colour space.
- HIGH_CONTRAST: We increase the contrast of each reference-image by scaling the mean-normalized pixel histograms $1.5\times$.
- LOW_CONTRAST: We decrease the contrast of each reference-image by scaling the mean-normalized pixel histograms $0.5\times$.
- JPEG_COMPRESS: We iteratively jpeg-compress the reference-image until its peak-signal-to-noise ratio is 26.

C. **Style**

- PENCIL_SKETCH: We regenerate images using the reference-prompts and replacing "high-resolution image/8k image" with "hand-drawn sketch/pencil drawing/black-and-white doodle" in them, followed by manual curation. The end data-type images look like pencil sketches of animals.

- CARTOON: We regenerate images using the reference-prompts and replacing "high-resolution image/8k image" with "cartoon drawing/cartoon/children's cartoon" in them, followed by manual curation. The end data-type images look like cartoon animals.
- EMBROIDERY: We regenerate images using the reference-prompts and replacing "high-resolution image/8k image" with "knitted embroidered pattern/knitted embroidery/embroidered shawl" in them, followed by manual curation. The end data-type images look like embroideries of animals.
- PLUSHIE: We regenerate images using the reference-prompts and replacing "high-resolution image/8k image" with "plushie" in them, followed by manual curation. The end data-type images look like plushie toys of animals.
- GRAFFITI: We regenerate images using the reference-prompts and replacing "high-resolution image/8k image" with "spray-painted graffiti/graffiti art/modern-style graffiti" in them, followed by manual curation. The end data-type images look like graffitis of animals.
- TATTOO: We regenerate images using the reference-prompts and replacing "high-resolution image/8k image" with "body tattoo/hand tattoo/leg tattoo" in them, followed by manual curation. The end data-type images look like tattoos of animals.
- SCULPTURE: We regenerate images using the reference-prompts and replacing "high-resolution image/8k image" with "marble/stone/bronze sculpture/statue" in them, followed by manual curation. The end data-type images look like animal sculptures.
- ORIGAMI: We regenerate images using the reference-prompts and replacing "high-resolution image/8k image" with "origami/origami toy/origami drawing" in them, followed by manual curation. The end data-type images look like origami animals.

D. **Semantic**

- TYPOGRAPHIC: We paste a random text at a random position on top of every reference-image.
- MULTI_SAME: We regenerate images using the reference-prompts and replacing "a {animal}" with 'pack/herd/group/team of {animals}" in them, followed by manual curation. The end data-type images contain multiple animals instead of a single one.
- MULTI_DIFFERENT: We use GLIGEN (Li et al., 2023d) to in-paint a tiger into each reference-image. The end data-type images contain two animals, a tiger and the original animal in the reference-image.
- TIGER_STRIPES: We regenerate images using the reference-prompts and add one of these prompts to the them: ["with the distinctive stripes of a (tiger:0.9) on its body", "displaying the unique stripes of a (tiger:0.9) on its face and limbs", "with the eye-catching stripes of a (tiger:0.9) on its body", "bearing the prominent stripes of a (tiger:0.9) on its face and limbs", "having the stunning and distinctive stripes of a (tiger:0.9) on its body", "with skin containing the characteristic stripes of a (tiger:0.9) on its face and limbs"], followed by manual curation. The end data-type images contain animals with tiger-striped skin or features on them.

## A.4 NATURALTYPEIDENT CONSTRUCTION

Here, we manually curated 50 reference-images from the KaggleAnimalImages dataset (Banerjee, 2023) across the same animal categories as before. For creating images for all data-types in the **geometric** and **pixel** categories, and the TYPOGRAPHIC data-type in the **semantic** category, we applied the same transformation functions detailed previously on the curated reference-images. For MULTI_DIFFERENT, we manually curated images from KaggleAnimalImages where there was more than a single animal in the image. For sourcing data-type images for the **style** categories, we followed a 3-step procedure: (1) we first searched google images with data-type specific animal-prompts, e.g., "a cartoon dog", "a graffiti of a lion" etc. We applied a time-filter on the search to only retrieve images post January 2022, to ensure that the retrieved images are not contained in LAION-2B-en, CLIP's pre-training dataset, (2) we manually curated 100 images per data-type across animals ensuring data quality and diversity, and (3) we de-duplicated our final set of 100 images per data-type against LAION-2B-en with an image indexed-search in LAION-5B and the PHash algorithm. This procedure resulted in a curated set of 50 images for all the **style** data-types. We how-

ever were unable to find an effective method for sourcing data-type images for MULTI_DIFFERENT and TIGER_STRIPES, hence our *NaturalTypeIdent* dataset only contains images from 25 data-types (1,250 samples).

# B DETAILS ABOUT MODEL EVALUATION

We first enlist all the tested models and then describe the individual evaluation strategies used for C-VLMs and LMMs.

## B.1 MODELS TESTED

A. **C-VLMs**

- **CLIP.** We tested 9 models in total using the OpenCLIP repository: CLIP-ResNet50-openai, CLIP-ResNet101-openai, CLIP-ViT-B-32-openai, CLIP-ViT-B-32-laion2b, CLIP-ViT-B-16-2b, CLIP-ViT-L-14-2b, CLIP-ViT-H-14-2b, CLIP-ViT-g-14-2b and CLIP-ViT-bigG-14-2b.
- **CoCa.** We tested 2 models using the OpenCLIP repository: CoCa-ViT-B-32-laion-2b and CoCa-ViT-L-14-2b.
- **BLIP-2-ITM.** We tested 2 "blip2_image_text_matching" models using the LAVIS repository: pretrain and pretrain_vitL

B. **LMMs**

- **Fromage.** We used the open-source implementation provided by the authors.
- **Multimodal-GPT.** We used the open-source implementation provided by the authors.
- **OpenFlamingo.** We used the OpenFlamingo-9B-HF model released with the Otter repository.
- **Otter.** We tested 2 models using the Otter repository: OTTER-Image-LLaMA7B-LA-InContext and luodian/OTTER-Image-MPT7B.
- **GILL.** We used the open-source implementation provided by the authors.
- **MPlugOwl.** We tested 4 models using the MPlugOwl open-source implementation: mplug-owl-llama-7b-pt, mplug-owl-llama-7b, mplug-owl-llama-7b-ft and mplug-owl-bloomz-7b-multilingual.
- **LLaVA.** We tested 3 models using the LLaVA open-source implementation: LLaVA-7B-v0, LLaVA-13B-v0 and LLaVA-Lightning-MPT-7B-preview.
- **BLIP-2-LLM.** We tested 3 "blip2_t5" models: pretrain_flant5xl, pretrain_flant5xxl and pretrain_flant5xl_vitL, and 2 "blip2_opt" models: pretrain_opt2.7b and pretrain_opt6.7b, using the LAVIS repository.
- **InstructBLIP.** We tested 2 "blip2_t5_instruct" models: flant5xl and flant5xxl, and 2 "blip2_vicuna_instruct" models: vicuna7b and vicuna13b, using the LAVIS repository.
- **IDEFICS.** We tested 4 models using the open-source huggingface release by the authors: idefics-9b, idefics-9b-instruct, idefics-80b and idefics-80b-instruct

## B.2 DATA-TYPE TEXT-PROMPTS USED FOR EVALUATION

For evaluating C-VLMs by cosine-similarity scoring, and LMMs by log-likelihood scoring, we used a default, fixed set of 27 data-type text prompts as detailed below:

- GAUSSIAN_NOISE: "This is a noisy image of an animal."
- DEFOCUS_BLUR: "This is a blurred image of an animal."
- SNOW: "This is a snowy image of an animal."
- HIGH_BRIGHTNESS: "This is a bright image of an animal."
- LOW_BRIGHTNESS: "This is a dark image of an animal."
- HIGH_CONTRAST: "This is a high contrast image of an animal."
- LOW_CONTRAST: "This is a low contrast image of an animal."

- JPEG_COMPRESS: "This is a jpeg compressed image of an animal."
- LEFT_ROTATE: "This is a left-rotated image of an animal."
- RIGHT_ROTATE: "This is a right-rotated image of an animal."
- PATCH_AND_RESHUFFLE: "This is a patched and reshuffled image of an animal."
- CROP_AND_ZOOM: "This is a randomly cropped and zoomed image of an animal."
- VERTICAL_FLIP: "This is a vertically flipped image of an animal."
- MIXUP: "This is an image of an animal mixed with another image."
- CUTMIX: "This is an image of an animal with one patch replaced by another image."
- PENCIL_SKETCH: "This is a pencil sketch of an animal."
- CARTOON: "This is a cartoon animal."
- EMBROIDERY: "This is an embroidered animal."
- PLUSHIE: "This is a plushie animal."
- GRAFFITI: "This is a graffiti of an animal."
- TATTOO: "This is a tattoo of an animal."
- SCULPTURE: "This is a sculpture of an animal."
- ORIGAMI: "This is an origami animal."
- TYPOGRAPHIC: "This is an image of an animal with some text written on top."
- MULTI_SAME: "This is an image of multiple animals."
- MULTI_DIFFERENT: "This is an image of a tiger and an animal."
- TIGER_STRIPES: "This is an image of an animal with tiger stripes."

### B.3 DATA-TYPE PROMPT-VARIATIONS FOR TESTING EFFECTS OF DIFFERENT PROMPTS ON RESULTS

To further test how much of an effect different prompting styles have on our results, we use two alternate prompt variations for testing our models too. We enlist the alternate sets of prompts used here. The first prompt variation list is:

- GAUSSIAN_NOISE: "This is a grainy image of an animal."
- DEFOCUS_BLUR: "This is a blurry image of an animal."
- SNOW: "This is an image of an animal in the snow."
- HIGH_BRIGHTNESS: "This image of an animal is bright."
- LOW_BRIGHTNESS: "This image of an animal is dim."
- HIGH_CONTRAST: "The contrast in this animal image is high."
- LOW_CONTRAST: "The contrast in this animal image is low."
- JPEG_COMPRESS: "This is an image of an animal in jpeg format."
- LEFT_ROTATE: "This is a counter-clockwise rotated image of an animal."
- RIGHT_ROTATE: "This is a clockwise rotated image of an animal."
- PATCH_AND_RESHUFFLE: "This is a scrambled image of an animal."
- CROP_AND_ZOOM: "This image is a close-up crop of an animal."
- VERTICAL_FLIP: "This is an upside-down image of an animal."
- MIXUP: "This is a mixed image of two animals."
- CUTMIX: "This is an image of an animal with a patch swapped from another image."
- PENCIL_SKETCH: "This is a hand-drawn sketch of an animal."
- CARTOON: "This is an animated animal."
- EMBROIDERY: "This looks like animal embroidery."

- PLUSHIE: "This is a stuffed toy animal."
- GRAFFITI: "This is graffiti art of an animal."
- TATTOO: "This is an inked image of an animal on skin."
- SCULPTURE: "This is a statue of an animal."
- ORIGAMI: "This is a paper-folded (origami) animal."
- TYPOGRAPHIC: "This is an image of an animal with some text overlaid."
- MULTI_SAME: "This image shows several of the same animals."
- MULTI_DIFFERENT: "This image shows both a tiger and another type of animal."
- TIGER_STRIPES: "This animal seems to have the pattern of tiger stripes."

The second prompt variation list is:

- GAUSSIAN_NOISE: "This is an image of an animal with some graininess."
- DEFOCUS_BLUR: "This is an out-of-focus image of an animal."
- SNOW: "This is an image of an animal seen amidst a snowy backdrop."
- HIGH_BRIGHTNESS: "This is an image of an animal, the image is bright."
- LOW_BRIGHTNESS: "This is an image of an animal, the image is dark."
- HIGH_CONTRAST: "This is an image of an animal, the image has high contrast."
- LOW_CONTRAST: "This is an image of an animal, the image has low contrast."
- JPEG_COMPRESS: "This is a compressed image of an animal."
- LEFT_ROTATE: "This is an image of an animal, the image is rotated 90 degrees to the left."
- RIGHT_ROTATE: "This is an image of an animal, the image is rotated 90 degrees to the right."
- PATCH_AND_RESHUFFLE: "This is a jumbled image of an animal."
- CROP_AND_ZOOM: "This is an image of an animal, the image is cropped and resized."
- VERTICAL_FLIP: "This is an image of an animal, the image is flipped upside-down."
- MIXUP: "This is an image of an animal with mixup augmentation."
- CUTMIX: "This is an image of an animal with cutmix augmentation."
- PENCIL_SKETCH: "This is a sketch of an animal."
- CARTOON: "This is a rendition of an animal in a cartoon style."
- EMBROIDERY: "This is an animal embroidery piece."
- PLUSHIE: "This is a soft toy animal."
- GRAFFITI: "This is an image of animal street art."
- TATTOO: "This is an image of an animal in tattoo style."
- SCULPTURE: "This is an image of an animal, the image looks like a sculpture."
- ORIGAMI: "This is an image of an animal crafted out of paper folding (origami)."
- TYPOGRAPHIC: "This is an image of an animal, the image has some text superimposed."
- MULTI_SAME: "This is an image of multiple same animals."
- MULTI_DIFFERENT: "This is an image of an animal and a tiger."
- TIGER_STRIPES: "This is an image of an animal patterned with tiger-like markings."

### B.4 DETAILS ON LOG-LIKELIHOOD SCORING FOR LMMS

To compute the zero-shot performance of C-VLMs, we computed the cosine similarity matching score of each image $I$ with each of the 27 data-type text prompts $\{D_1, D_2, \ldots, D_{27}\}$ (as enumerated in Appendix B.2) and predicted the data type with the highest score, i.e., `predicted_data_type`=$\arg\max_{i\in\{1,\ldots,27\}}$ `cos_sim`(`I_enc`$(I)$, `T_enc`$(D_i)$). Since LMMs are auto-regressively trained, evaluating them in the same way is impossible. For the most fair comparison of both model classes, we instead compute log-likelihoods, of the prompt containing the image with the appended data-type text prompts $D_i$ to the image. Specifically, for each image $I$, we compute the log-likelihood under the LMM, for the query prompt $P\_i$="<image> Q: Describe the image. A: <D_i>", where <D_i> is replaced by the specific data-type text prompt. More concretely, for a particular data-type prompt $P\_i$, we tokenize the prompt $P\_i$, pass these input tokens into the model, and retrieve the logits for each of the input tokens. Then, we sum up the log-probabilities of each input token, and use that as the final aggregated log-probability $L\_i$ for the particular data-type. We repeat this process for all data-types, and then predict the data-type with the highest log-probability i.e., `predicted_data_type`=$\arg\max_{i\in\{1,\ldots,27\}} L\_i$. Note that the default query prompt $P$ differs across different models depending on the particular prompts used for instruction tuning, and we match our prompting strategy accordingly (see Appendix B.5).

This log-likelihood evaluation strategy is common for evaluating LLMs on multiple choice questions (Gao, 2023; Brown et al., 2020; Sanh et al., 2022) and LMMs on classification tasks (Dai et al., 2023; Awadalla et al., 2023). Further, as enumerated in (Gao, 2023), we tried different length normalisation strategies while computing the log-likelihood scores, but observed no significant changes to the results—hence we used the standard log-likelihood scoring procedure outlined above.

For concreteness, we also showcase the exact code snippets for computing these log-probabilities for three models, LLaVA, IDEFICS and Otter.

```
1
2 '''
3 The below functions for computing log-likelihoods are defined within a model class where we
       have defined a model, tokenizer and other auxiliary variables. The model itself is an
       instance of LlavaLlamaForCausalLM, defined as:
4
5 self.tokenizer = AutoTokenizer.from_pretrained(model_name)
6 self.model = LlavaLlamaForCausalLM.from_pretrained(model_name, low_cpu_mem_usage=True,
       torch_dtype=torch.float16, use_cache=True).cuda()
7 self.image_processor = CLIPImageProcessor.from_pretrained(self.model.config.mm_vision_tower,
       torch_dtype=torch.float16)
8 '''
9
10     def log_lik_scores(self, images, prompt, option):
11         '''
12         Compute log-likelihood score under the model given an input image, a text prompt, and
       an answer option
13         '''
14
15         qs = prompt
16
17         if self.mm_use_im_start_end:
18             qs = qs + '\n' + DEFAULT_IM_START_TOKEN + DEFAULT_IMAGE_PATCH_TOKEN * self.
       image_token_len + DEFAULT_IM_END_TOKEN
19         else:
20             qs = qs + '\n' + DEFAULT_IMAGE_PATCH_TOKEN * self.image_token_len
21
22         if "v1" in self.pretrained.lower():
23             conv_mode = "llava_v1"
24         elif "mpt" in self.pretrained.lower():
25             conv_mode = "mpt_multimodal"
26         else:
27             conv_mode = "multimodal"
28
29         conv = conv_templates[conv_mode].copy()
30         conv.append_message(conv.roles[0], qs)
31         conv.append_message(conv.roles[1], None)
32         # This is the final prompt that we use as the default text prompt to probe with.
33         prompt = conv.get_prompt()
34
35         image_tensor = self.image_processor.preprocess(images, return_tensors='pt')['
       pixel_values'][0]
36
37         target_prompt = prompt + ' {}'.format(option)
```

```
38
39          inputs = self.tokenizer([target_prompt])
40
41          input_ids = torch.as_tensor(inputs.input_ids).cuda()
42          attention_mask = torch.as_tensor(inputs.attention_mask).cuda()
43
44          with torch.inference_mode():
45              outputs = self.model.forward(
46                  input_ids=input_ids,
47                  labels=input_ids,
48                  attention_mask=attention_mask,
49                  images=image_tensor.unsqueeze(0).half().cuda(),
50                  )
51
52          return -outputs.loss.item()
53
54      def get_prediction_from_model(self, image, text_classes):
55          '''
56          Zero-shot classify the image with the given text class transformed prompts
57          '''
58
59          log_scores = []
60
61          prompt_to_use = 'Describe the image.'
62
63          for gt_p in text_classes:
64              input_images = image
65              input_prompt = prompt_to_use
66              model_out = log_lik_scores(input_images, input_prompt, gt_p)
67              log_scores.append(model_out)
68
69          pred = np.argmax(log_scores, axis=-1)
70
71          return pred
```

Listing 1: LLaVA log-likelihood scoring

```
1
2   '''
3   The below functions for computing log-likelihoods are defined within a model class where we
        have defined a model, tokenizer and other auxiliary variables. The model itself is an
        instance of IdeficsForVisionText2Text, defined as:
4
5   self.model = IdeficsForVisionText2Text.from_pretrained(checkpoint, torch_dtype=torch.bfloat16,
         low_cpu_mem_usage=True, device_map='auto')
6   self.processor = AutoProcessor.from_pretrained(checkpoint)
7   '''
8
9       def log_lik_scores(self, context_inputs, prompt, option):
10          """
11          Compute log-likelihood score under the model given an input image, a text prompt, and
        an answer option
12          """
13
14          context_with_answer_inputs = self.processor(prompt[:-1] + [prompt[-1] + ' {}'.format(
        option)], return_tensors="pt", debug=False).to(self.device)
15          context_with_answer_inputs['labels'] = context_with_answer_inputs['input_ids'].clone()
        .to(self.device)
16          loss = self.model(**context_with_answer_inputs).loss.float().item()
17          return -loss
18
19      def get_prediction_from_model(self, image, text_classes):
20          log_scores = []
21
22          prompt_to_use = 'Describe the image.'
23          prompt_to_use = [
24              "User:",
25              image,
26              "{}\nAssistant:".format(prompt_to_use),
27          ]
28
29          input_images = image
30          context_prompt = prompt_to_use
31
32          context_inputs = self.processor(context_prompt, return_tensors="pt", debug=False).to(
        self.device)
33
34          for gt_p in text_classes:
35              model_out = self.log_lik_scores(context_inputs, prompt_to_use, gt_p)
36              log_scores.append(model_out)
```

```
37
38          pred = np.argmax(log_scores, axis=-1)
39          return pred
```

Listing 2: IDEFICS log-likelihood scoring

```
1
2  '''
3  The below functions for computing log-likelihoods are defined within a model class where we
       have defined a model, tokenizer and other auxiliary variables. The model itself is an
       instance of OtterForConditionalGeneration, defined as:
4
5  self.model = OtterForConditionalGeneration.from_pretrained(PRETRAINED[self.model_pretrained],
       device_map="auto")
6
7  self.tokenizer = self.model.text_tokenizer
8  self.image_processor = transformers.CLIPImageProcessor()
9  self.model.eval()
10 self.model.cuda()
11 '''
12
13     def find_sub_list(self, sl, l):
14         """
15         Utility function to find sub-list in a given list, used for
16         computing likelihood of answer tokens only without computing
17         likelihood of prompt context tokens
18         """
19         results = []
20         sll = len(sl)
21         for ind in (i for i, e in enumerate(l) if e == sl[0]):
22             if l[ind : ind + sll] == sl:
23                 results.append(ind + sll - 1)
24         return results
25
26     def log_lik_scores(self, vision_x, prompt, option, prompt_tokens):
27         """
28         Compute log-likelihood score under the model given an input image, a text prompt, and
       an answer option
29         """
30
31         target_text = f"{prompt} {option}"
32
33         lang_x = self.model.text_tokenizer([target_text], return_tensors="pt")
34
35         outputs = self.model(
36             vision_x=None,
37             lang_x=lang_x["input_ids"].cuda(),
38             attention_mask=lang_x["attention_mask"].cuda(),
39             clear_conditioned_layers=False,
40             use_cached_vision_x=True,
41         )
42
43         probs = torch.softmax(outputs.logits, dim=-1).detach()
44         probs = probs[:, :-1, :]
45         input_ids = lang_x["input_ids"][:, 1:].cuda()
46         gen_probs = torch.gather(probs, 2, input_ids[:, :, None]).squeeze(-1)
47
48         probs = []
49         for input_sentence, input_probs in zip(input_ids, gen_probs):
50             idxes = self.find_sub_list(prompt_tokens, input_sentence.detach().cpu().numpy().
       tolist())
51             input_probs = input_probs[idxes[-1] + 1 :]
52             probs.append(torch.prod(input_probs).item())
53
54         return probs[0]
55
56     def get_prediction_from_model(self, image, text_classes):
57         log_scores = []
58
59         prompt_to_use = "<image> Q: Describe the image. A:"
60
61         input_images = [image]
62
63         media_token_id = self.tokenizer("<image>", add_special_tokens=False)["input_ids"][-1]
64
65         vision_x = (
66             self.image_processor.preprocess(input_images, return_tensors="pt")["pixel_values"
       ].unsqueeze(1).unsqueeze(0)
67         )
68         # pre-cache the vision features for current sample here
```

```
69        self.model._encode_vision_x(vision_x.cuda())
70
71        prompt_tokens = (
72            self.tokenizer(prompt_to_use, add_special_tokens=False, return_tensors="np")["
    input_ids"].ravel().tolist()
73        )
74
75        for gt_p in text_classes:
76            model_out = self.log_lik_scores(vision_x, prompt_to_use, gt_p, prompt_tokens)
77            log_scores.append(model_out)
78
79        pred = np.argmax(log_scores, axis=-1)
80        return pred
```

Listing 3: Otter log-likelihood scoring

### B.5 LMM-SPECIFIC EVALUATION DETAILS

By default, for a fair comparison with C-VLMs, we evaluate LMMs using log-likelihood scoring with the prompt: "`<image> Q: Describe the image. A: <data_type_description>`", where `data_type_description` is substituted with each of the prompts from the section above. However, some LMMs have specific prompting styles, which we incorporate to ensure correct evaluation, as follows:

- **Multimodal-GPT.**

  <BOS>Below is an instruction that describes a task. Write a response that appropriately completes the request.
  ### Image: <image>
  ### Instruction: Describe the image.
  ### Response:

- **MPlugOwl.**

  The following is a conversation between a curious human and AI assistant. The assistant gives helpful, detailed, and polite answers to the user's questions.
  Human: <image>
  Human: Describe the image.
  AI:

- **LLaVA-7/13B-v0**

  You are LLaVA, a large language and vision assistant trained by UW Madison WAIV Lab.You are able to understand the visual content that the user provides, and assist the user with a variety of tasks using natural language.Follow the instructions carefully and explain your answers in detail.
  ###Human: Hi!
  ###Assistant: Hi there! How can I help you today?
  ###Human: Describe the image.<im_start><im_patch><im_end>
  ###Assistant:

- **LLaVA-MPT-7B**

| Prompt variation | Averaged-mean informedness across models |
|---|---|
| Default | 0.228 |
| Alternate 1 | 0.191 |
| Alternate 2 | 0.197 |

Table 4: **The default prompts used obtain the best average performance across models.**

---

<—**im_start**—>
**system**:
- You are LLaVA, a large language and vision assistant trained by UW Madison WAIV Lab.
- You are able to understand the visual content that the user provides, and assist the user with a variety of tasks using natural language.
- You should follow the instructions carefully and explain your answers in detail.

<—**im_end**—>
<—**im_start**—>
**user**: Describe the image.
<—**im_start**—><—**im_patch**—><—**im_end**—>
**assistant**:

---

## C PROMPT VARIATION EXPERIMENTS

For our main zero-shot probing results in Sec. 4.2, we used a set of default, fixed prompts for each data-type (as listed in Appendix B.2). To further test if our results are sensitive to variations in different prompting styles, we also ran prompt variation experiments on a subset of our models with two alternative sets of data-type prompts (as listed in Appendix B.3). In Fig. 7, we showcase the results with the three different prompting styles—we observe largely the same qualitative trends across the three different prompting styles, with the C-VLMs still outperforming LMMs. We do however note that the absolute averaged mean-informedness values across models is quite different (see Tab. 4) suggesting that different prompting styles do make a slight difference for the absolute mean informedness values, however the overall results (C-VLMs outperforming LMMs, LMMs performing generally poor, only marginal scaling behaviour) remain the same.

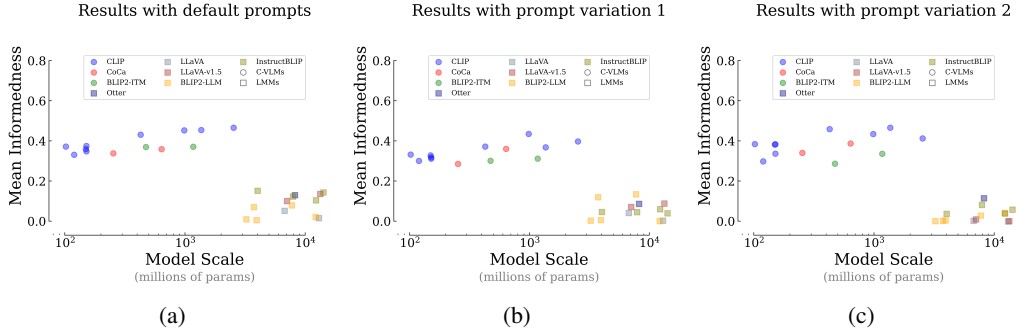

Figure 7: **Prompt sensitivity for zero-shot results**. The overall qualitative trends remain the same even with different prompt variations for testing the zero-shot models, for both C-VLMs and LMMs.

Despite showcasing results with three different prompt variations, it is still inconclusive if we are using the optimal set of prompts for probing the VLMs. To further extend this analysis, we use the method from VisDesc (Menon & Vondrick, 2022) to generate a large-set of descriptors per data-type by prompting GPT-3 (Brown et al., 2020) with the following:

> Q: What are useful visual features for distinguishing a lemur in a photo?
> A: There are several useful visual features to tell there is a lemur in a photo:
> - four-limbed primate
> - black, grey, white, brown, or red-brown
> - wet and hairless nose with curved nostrils
> - long tail
> - large eyes
> - furry bodies
> - clawed hands and feet
>
> Q: What are useful visual features for distinguishing a {category name} in a photo?
> A: There are several useful visual features to tell there is {category name} in a photo:
> -

where {category_name} is replaced by a description of each of our data-types.

We then use these several descriptor prompts for our zero-shot analysis. For probing C-VLMs, we directly average the text embeddings corresponding to the different text descriptors per-data-type, and compute similarities with the test image (exactly the method used in VisDesc). Similarly, for LMMs, we compute the log-likelihoods of each of the text descriptors per-data-type and average them. We then take the data-type with the maximum averaged log-likelihood as the prediction for that test sample[1].

## Results with visual descriptions

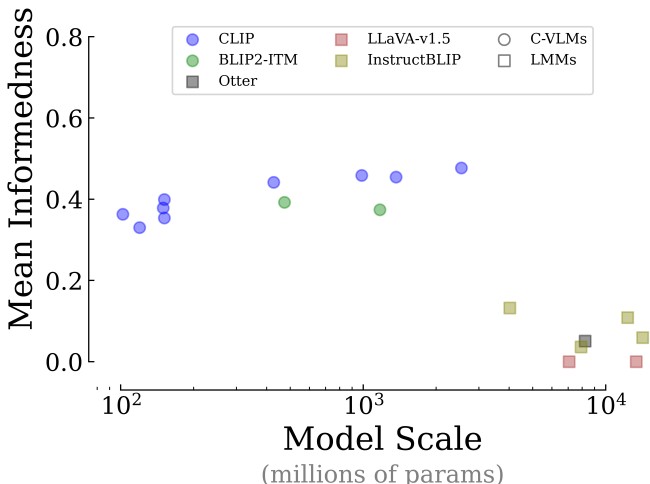

Figure 8: **Results with visual descriptor prompts from VisDesc**

Fig. 8 shows our zero-shot results with these enhanced descriptor prompts. With this probing method, we again do not see any significant deviations in trends from the default probing prompts we used. We therefore conclude that despite different prompts having some effect on the final results, our language-based zero-shot probing results are fairly robust across prompting methods.

## D  ANIMAL IDENTIFICATION EXPERIMENTS

To ensure that the results we see in Fig. 3 are indeed informative and not an artefact of incorrect evaluation, we run a control experiment where we tested models on identifying animals in *SyntheticTypeIdent*. Due to time-constraints, we omit the IDEFICS models from this analysis. In Fig. 9,

---

[1]This log-likelihood averaging has also been used previously for evaluating OpenFlamingo

we plot the mean informedness across the 10 animal classes, for all models. Compared to our main results on data-type identification ( Fig. 3), all models, including the LMMs, are extremely good at identifying the animal in the images. This suggests that VLMs are more adept at discriminating the semantic concepts within an image over identifying data-types. This result also connects with our embedding space analysis in Fig. 5.

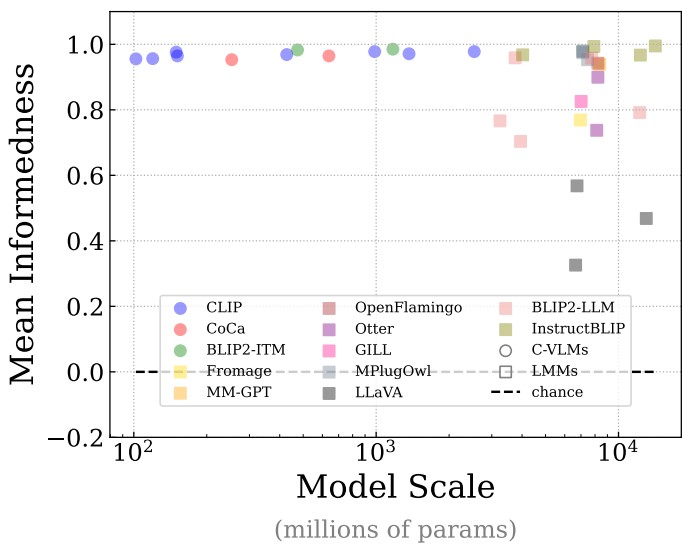

Figure 9: **Performance in identifying animals on *SyntheticTypeIdent*.**

## E   DISCUSSION ON DATA-AUGMENTATION AND CONNECTIONS TO DATA-TYPE IDENTIFICATION

Data-augmentation is a standard technique used for improving generalisation (Raileanu et al., 2021; Mikołajczyk & Grochowski, 2018; Perez & Wang, 2017), reducing overfitting (Taylor & Nitschke, 2018; Shorten & Khoshgoftaar, 2019), and enhancing robustness to distribution shifts (Rebuffi et al., 2021b; Zhong et al., 2020; Zhao et al., 2020; Rebuffi et al., 2021a), in deep neural networks. Almost all the ImageNet-trained CNNs of the last decade (Krizhevsky et al., 2012; Simonyan & Zisserman, 2014; Zeiler & Fergus, 2014; He et al., 2016; Szegedy et al., 2015) have used some form of data-augmentation for improving their generalisation performance. Further, the fields of self-supervised (Jaiswal et al., 2020) and semi-supervised (Van Engelen & Hoos, 2020) visual representation learning almost solely rely on data-augmentation techniques for capturing their supervision signals. A prime example, SimCLR (Chen et al., 2020), relies on using simple data-augmentation techniques like flipping, rotating or cropping for procuring its positive image samples for the instance-based contrastive loss. Therefore, these networks rely on an *invariance prior* that makes them insensitive to capturing data-type information. Hence, we can expect them to underperform in identifying data-types since the information is never captured in the embeddings by account of this *invariance prior*. However, we note that CLIP does not use any data-augmentation while training other than random-resized cropping. Similarly, BLIP-2 and CoCa only use random resized cropping and horizontal flipping (not a part of our data-types) as augmentations. Therefore, this suggests that despite models not being explicitly subjected to learning invariances, they learn to be invariant to understanding data-types (as we show in Fig. 5). We further showed in Sec. 5 that this emergent insensitivity to data-types can be traced back to the lack of data-type rich pre-training data. Therefore, our current models—whether explicitly or implicitly—do not capture information required for distinguishing data-types, and we make a strong case why this is important for the real-world. Through our *data-type identification* framework, we call for rethinking the current practices of training VLMs that induce invariances in them, and show why it is an important practical problem.

| k | Animal-class | Data-type |
|---|---|---|
| 1 | 0.96 | 0.29 |
| 5 | 0.96 | 0.30 |
| 10 | 0.94 | 0.33 |
| 50 | 0.97 | 0.40 |

Table 5: **K-NN results on *SyntheticTypeIdent***. We present the mean informedness of k-nearest-neighbours classifiers on the SyntheticTypeIdent dataset for predicting the animal-class or the data-type for each of our samples based on CLIP's image-feature embeddings.

## F  QUANTIFYING LINEAR SEPARABILITY OF ANIMAL- AND DATA-TYPES IN CLIP'S IMAGE-SPACE

To quantify how invariant CLIP-RN50's vision encoder is to identifying animal-classes and data-types (see Fig. 5), we run k-nearest-neighbours experiments on the CLIP-RN50 image embeddings as features and animal-classes / data-types as labels. While the KNN classifier performed quite well on identifying the animal in the image, performance significantly dropped in identifying the data-type (see Tab. 5). Similarly, a linear probe (multinomial logistic regression with LBFGS optimizer akin to what was used for CLIP-RN50's original linear probing results (Radford et al., 2021)) showed high linear separability of animal classes (mean informedness=0.99), whereas data-types were less linearly separable (mean informedness=0.84). These values were obtained on the train set only, as we want to study linear separability of the data points and are not interested in test set generalisation. The number of 1,350 data-points in our dataset is similar to the dimensionality of CLIP-RN50's feature space (1,024 dimensions), which might in part account for the higher performance of the logistic regression classifier, confounding our results slightly. Overall, the worse performance of these classifiers when applied on data-types compared to animal-types suggest CLIP is somewhat invariant to data-types compared to identifying animal classes.

## G  ANALYSING THE PRE-TRAINING DISTRIBUTION

For conducting the analysis in Sec. 5, we first searched the entire LAION-2B-en text index for keyword-hits for each data-type. We consider a hit if a keyword (or its associated term) is verbatim present in the text caption of the candidate sample's text caption. We here enumerate the different search keywords we used for this analysis for all data-types. The vertical bar ("|") denotes a logical OR in regex syntax, indicating any one of the keywrods separated by "|" can be matched.

- GAUSSIAN_NOISE: "noisy"
- DEFOCUS_BLUR: "blurred"
- SNOW: "snowy"
- HIGH_BRIGHTNESS: "bright"
- LOW_BRIGHTNESS: "dark"
- HIGH_CONTRAST: "high contrast"
- LOW_CONTRAST: "low contrast"
- JPEG_COMPRESS: "jpeg|jpeg compressed|jpeg-compressed".
- LEFT_ROTATE: "left-rotated|left rotated"
- RIGHT_ROTATE: "right-rotated|right rotated"
- PATCH_AND_RESHUFFLE: "patched and reshuffled|patched & reshuffled|patched"
- CROP_AND_ZOOM: "cropped and zoomed|cropped & zoomed"
- VERTICAL_FLIP: "vertically flipped|flipped"
- MIXUP: "mixed with|mixup"
- CUTMIX: "patch replaced|cutmix"

- PENCIL_SKETCH: "pencil sketch"
- CARTOON: "cartoon"
- EMBROIDERY: "embroidered"
- PLUSHIE: "plushie"
- GRAFFITI: "graffiti"
- TATTOO: "tattoo"
- SCULPTURE: "sculpture"
- ORIGAMI: "origami"
- TYPOGRAPHIC: "text written"
- MULTI_SAME: "multiple"
- MULTI_DIFFERENT: "tiger and a"
- TIGER_STRIPES: "with tiger stripes"

We also provide the raw values of (1) the text-frequency in LAION-2B-en, (2) the alignment probability, and (3) the abundancy score, obtained for each data-type in Tab. 6. Note that since we compute the Spearman rank correlations between the informedness values per data-type and the abundancy score, the scales of the scores do not matter. We further provide the complete set of rank-correlations obtained between the abundancy scores and the performance (mean informedness) across all models as well as only across CLIP models, in Tab. 7. Finally, in Fig. 10 we provide qualitative examples of retrieved samples for different data-types, sorted by average model performance on that particular data-type (from Fig. 4) i.e., average model-performance decreases from left to right—a careful visual inspection gives us a sense of the strong association we see from the correlation scores—the more performant a particular data-type is, the more aligned the retrieved samples are to the data-type concept.

Table 6: **Raw text-frequency, alignment probability and abundancy scores across data-types.**

| Category | Data-Type | Text-frequency | Alignment probability | Abundancy score |
|---|---|---|---|---|
| **Geometric** | LEFT_ROTATE | $8.62 \times 10^{-9}$ | 0.294 | $2.54 \times 10^{-9}$ |
| | RIGHT_ROTATE | $1.16 \times 10^{-8}$ | 0.375 | $4.36 \times 10^{-9}$ |
| | PATCH_AND_RESHUFFLE | $5.04 \times 10^{-5}$ | 0.040 | $2.04 \times 10^{-6}$ |
| | CROP_AND_ZOOM | $4.31 \times 10^{-9}$ | 1.000 | $4.31 \times 10^{-9}$ |
| | VERTICAL_FLIP | $1.95 \times 10^{-5}$ | 0.141 | $2.75 \times 10^{-6}$ |
| | MIXUP | $2.98 \times 10^{-5}$ | 0.020 | $6.02 \times 10^{-7}$ |
| | CUTMIX | $1.59 \times 10^{-8}$ | 0.286 | $4.56 \times 10^{-9}$ |
| **Pixel** | GAUSSIAN_NOISE | $3.01 \times 10^{-5}$ | 0.131 | $3.95 \times 10^{-6}$ |
| | DEFOCUS_BLUR | $2.37 \times 10^{-4}$ | 0.810 | $1.92 \times 10^{-4}$ |
| | SNOW | $2.73 \times 10^{-4}$ | 0.677 | $1.85 \times 10^{-4}$ |
| | HIGH_BRIGHTNESS | $2.36 \times 10^{-3}$ | 0.649 | $1.53 \times 10^{-3}$ |
| | LOW_BRIGHTNESS | $4.68 \times 10^{-3}$ | 0.310 | $1.45 \times 10^{-3}$ |
| | HIGH_CONTRAST | $1.18 \times 10^{-5}$ | 0.570 | $6.72 \times 10^{-6}$ |
| | LOW_CONTRAST | $1.77 \times 10^{-6}$ | 0.454 | $8.05 \times 10^{-7}$ |
| | JPEG_COMPRESS | $1.87 \times 10^{-4}$ | 0.828 | $1.55 \times 10^{-4}$ |
| **Style** | PENCIL_SKETCH | $1.85 \times 10^{-5}$ | 0.950 | $1.76 \times 10^{-5}$ |
| | CARTOON | $2.45 \times 10^{-3}$ | 0.622 | $1.53 \times 10^{-3}$ |
| | EMBROIDERY | $1.97 \times 10^{-3}$ | 0.697 | $1.37 \times 10^{-3}$ |
| | PLUSHIE | $3.07 \times 10^{-5}$ | 0.750 | $2.30 \times 10^{-5}$ |
| | GRAFFITI | $2.50 \times 10^{-4}$ | 0.617 | $1.54 \times 10^{-4}$ |
| | TATTOO | $1.43 \times 10^{-3}$ | 0.515 | $7.38 \times 10^{-4}$ |
| | SCULPTURE | $6.64 \times 10^{-4}$ | 0.745 | $4.95 \times 10^{-4}$ |
| | ORIGAMI | $2.68 \times 10^{-4}$ | 0.727 | $1.95 \times 10^{-4}$ |
| **Semantic** | TYPOGRAPHIC | $3.33 \times 10^{-6}$ | 0.030 | $1.01 \times 10^{-7}$ |
| | MULTI_SAME | $7.03 \times 10^{-4}$ | 0.490 | $3.44 \times 10^{-4}$ |
| | MULTI_DIFFERENT | $7.66 \times 10^{-7}$ | 0.245 | $1.88 \times 10^{-7}$ |
| | TIGER_STRIPES | $1.29 \times 10^{-7}$ | 0.823 | $1.07 \times 10^{-7}$ |

Table 7: **Abundancy-scores are strongly correlated with informedness.**

| Dataset | All models, $r=$ | CLIP models, $r=$ |
|---|---|---|
| *SyntheticTypeIdent* | 0.557 | 0.606 |
| *NaturalTypeIdent* | 0.489 | 0.415 |

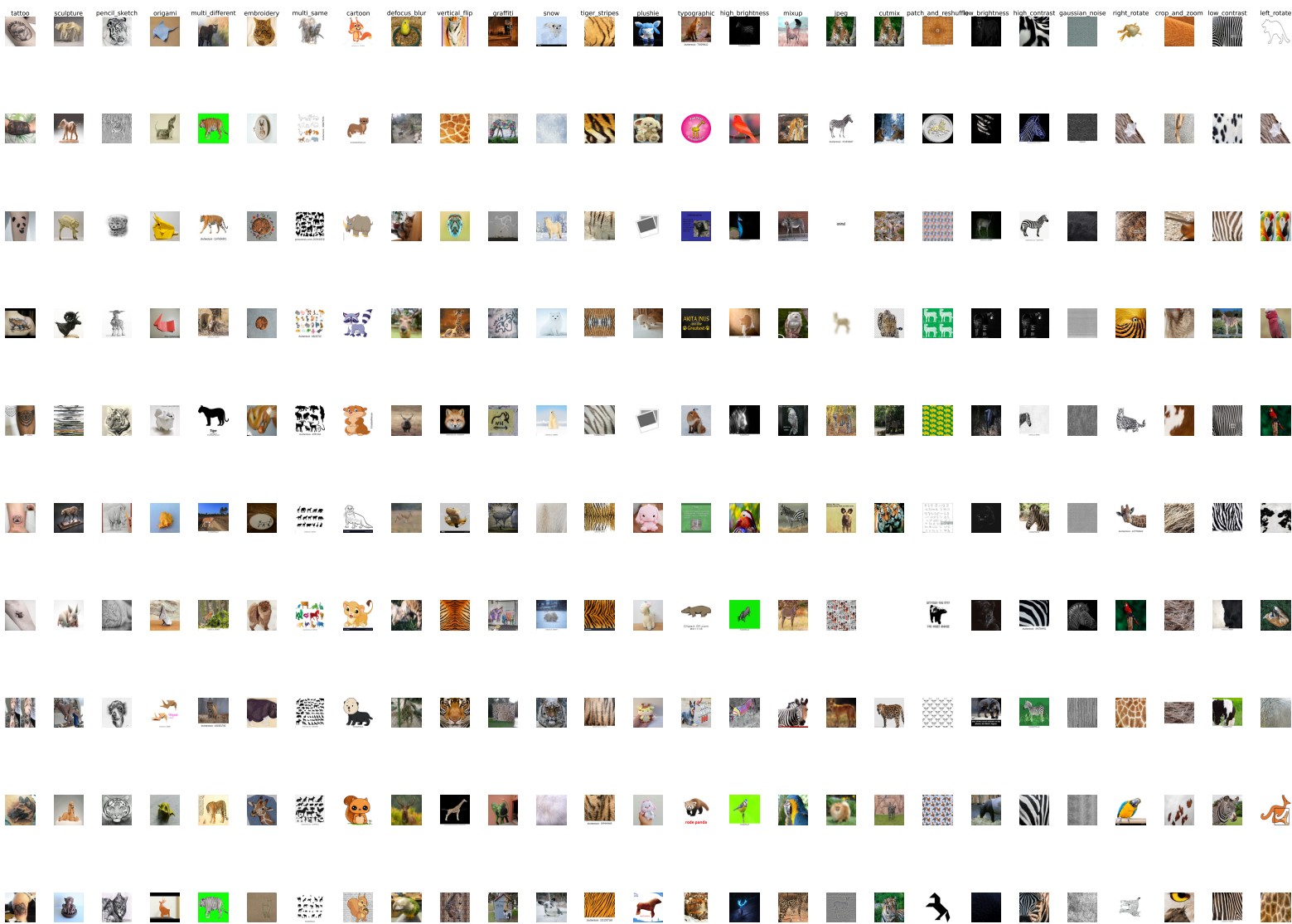

Figure 10: **Sample images per data-type from the retrieved set of correct data-type examples.**

### G.1 Replacing the text-based search with semantic-similarity search

For the main correlation analysis, we used a simple text-based search in the pre-training distribution's text captions. However, using semantic similarity metrics instead of a simple exact text-matching metric might help in creating a more comprehensive vocabulary for each data type (e.g., PATCH_AND_RESHUFFLE can also be represented by terms like collage, mosaic etc).

To probe this hypothesis, we used a semantic search based on the sentence-transformers/all-MiniLM-L6-v2 text embedding model. We make use of this model as it is both light-weight and has been optimised to work effectively for sentence similarity matching in large corpora (Reimers & Gurevych, 2019). Specifically, we encode the text descriptions of all 27 data-types using the all-MiniLM-L6-v2 model to get their embeddings. We also encode all the text captions in the LAION-2B-en pre-training dataset (note that due to compute constraints we only run this analysis on 36 out of the 127 parquet files available i.e., on approx. 30% of the pre-training data). For each data-type, we then match all the text captions from the LAION-2B-en dataset that have a cosine similarity larger than 0.85 with the data-type description embedding. We chose 0.85 as our matching threshold by manually inspecting similarity scores on a curated set of sentences containing data-type descriptions—below 0.85 we noted that the concepts within two sentences no longer matched. Once we obtain the matching pre-training samples per data-type, we follow the same procedure as in Sec. 5 to obtain *abundancy scores*. With this new semantic search method, we get a high rank correlation between *abundancy scores* and averaged model informedness, $r=0.798$ for *SyntheticTypeIdent*. For CLIP-only based model average informedness, the rank correlation is similarly very high: $r=0.776$. This result provides further evidence that the lack of such data-type rich samples in the pre-training dataset is a strong reason for the poor performance of VLMs on data-type identification.

## H Training-free adaptation experiments details

We augment the main text by providing more details about few-shot example selection for both methods, and an added interpretation of results here.

### H.1 CLIP Tip-Adapter

We used two strategies for selecting few-shot examples for adaptation: *Random*: for each test sample, we selected few-shot examples for each data-type randomly from across all the images from our test set for that data-type. Concretely, for a $n$-shot experiment, if a test sample is a left-rotated image of a dog, for each of the 27 data-types we randomly picked $n$ examples corresponding to that data-type. Since we have 10 animals and 5 examples per animal in our test set, we can pick the $n$ examples per data-type from 50 images for that data-type available in our test set, and (2) *SameAnimal*: for each test sample, we selected few-shot examples for each data-type such that all the few-shot examples contained the exact same animal as in the test sample. Concretely, if a test sample is a left-rotated image of a dog, for each of the 27 data-type we randomly picked $n$ examples corresponding to that data-type while ensuring that the picked samples always contain the same animal as the test sample. Since we have 10 animals and 5 examples per animal in our test set, we can pick the $n$ examples per data-type from 5 images for that data-type available in our test set (hence why our *SameAnimal* few-shot adaptation plot cuts off at 5 few-shot examples in Fig. 6a).

### H.2 Otter in-context learning

Motivated by previous works (Lu et al., 2021; Pezeshkpour & Hruschka, 2023) demonstrating several biases of in-context learning in LLMs, we experimented with multiple in-context example selection strategies. We present the two best performing strategies here: (1) *Random*: For selecting $n$ in-context examples for a test sample, we randomly select $n-1$ examples from the test set while always ensuring one in-context example that corresponds to the target data-type of the test sample. Concretely, if the test sample is a left-rotated image of a dog, we select $n-1$ random examples from the test set, and select one left-rotated image (the animal in the image need not be a dog) as an additional in-context example—this ensures that for a particular target data-type, Otter always has an example that describes that particular data-type, and (2) *SameAnimal*: For selecting $n$ in-context

examples for a test sample, we use the exact same procedure as the *Random* strategy, while additionally ensuring that each in-context example is an image of the same animal as the test sample. For our left-rotated dog example, we would ensure that all the selected in-context examples would be images of dogs.

## H.3 LLaVA in-context learning

We asked if in-context learning on our data-type identification task would improve with larger model sizes. Thus, we repeated the same experiments as for Otter (see Sec. 6.1, Fig. 6b, and Appendix H.2) for the larger LLaVA-13B and LLaVA-7B as a comparison. Interestingly, both model versions underperformed Otter-LLaMA7B and Otter-MPT7B, and surprisingly, LLaVA-13B even underperformed its smaller LLaVA-7B version (see Fig. 11).

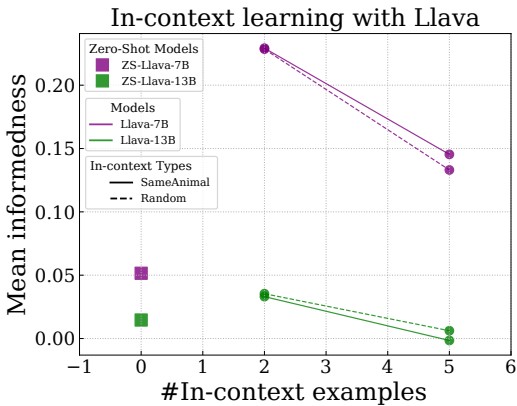

Figure 11: **In-context learning results for LLaVA-7B and LLaVA-13B.** On our data-type identification task, both models underperformed in-context learning with Otter, and the larger LLaVA model underperformed it's bigger variant.

## H.4 Explaining CLIP TIP-Adapter's failure

Why does providing few-shot target examples that should *prime* the model better for data-type understanding degrade performance over even the zero-shot model? We try to explain this a bit further here. TIP-Adapter utilises semantic similarities in CLIP's image embedding space for adaptation. However, as we've shown from Fig. 5, the image embedding space does not contain enough information to disambiguate between data-types, and hence the adaptation method is unable to capture any discriminative information for classifying between data-types; it rather exploits semantic similarities between concepts which is actually detrimental for our task. An intuitive example would be: say we are classifying an image of a noisy dog. The target data-type is GAUSSIAN_NOISE. For the adaptation, we would have picked random few-shot examples that contain the target data-type samples as well as other samples. Since TIP-Adapter utilises semantic similarity over matching data-types, it could assign higher image-image similarity scores for a data-type where all its few-shot examples are dogs, whereas the target data-type few-shot examples only contain other animals. This can therefore systematically degrade performance.

## I Details about TeDaTy construction

We constructed TeDaTy by sourcing training images from ImageNet (Deng et al., 2009), PACS (Li et al., 2017), and DomainNet (Peng et al., 2019). TeDaTy uses 8 in-distribution data-types (out of all 27) for fine-tuning, we enlist them here: GAUSSIAN_NOISE, DEFOCUS_BLUR, LEFT_ROTATE, RIGHT_ROTATE, PATCH_AND_RESHUFFLE, PENCIL_SKETCH, CARTOON and TYPOGRAPHIC. Our motivation for selecting these data-types was two-fold: (1) we wanted to have uniform representation of data-types across the four broad **geometric**, **pixel**, **style**, and **semantic** categories, and (2) we wanted to have a few data-types that models were already reason-

ably performant on (PENCIL_SKETCH, CARTOON) and others that the models completely fail on (GAUSSIAN_NOISE, PATCH_AND_RESHUFFLE).

Constructing the training samples for GAUSSIAN_NOISE, DEFOCUS_BLUR, LEFT_ROTATE, RIGHT_ROTATE, PATCH_AND_RESHUFFLE and TYPOGRAPHIC is straightforward—we first subsampled a 100k-sized subset (100 training samples per class) of ImageNet's training dataset (which we call IN100k), and then applied the transformation functions corresponding to each of the data-types accordingly. For CARTOON, we acquired training samples by combining the training datasets of PACS-cartoon and DomainNet-clipart. For PENCIL_SKETCH, we adapted the training dataset of DomainNet-sketch as is. Further, for each data-type transfored image, we ensured that the caption for that image incorporates the data-type information in it. For example, a CARTOON image of a dog acquired from the PACS-cartoon subset would have the caption as "This is a cartoon image of a dog.". Similarly, a LEFT_ROTATE image of a tench from the ImageNet subset would have the caption, "This is a left-rotated photo of a tench.". We provide a few sample visualisations of the specific training images along with their captions used in Fig. 12.

## J  FINE-TUNING DETAILS

### J.1  CLIP

For fine-tuning CLIP, we used the CLIP ViT-B-32 model for all experiments using the OpenCLIP repository. We used the standard contrastive loss for fine-tuning all our models for 5 epochs with the AdamW (Loshchilov & Hutter, 2017) optimiser, using 50 steps of warmup, cosine-annealing learning rate schedule and a batch-size of 512. For each experiment, we swept over 5 different learning-rates $\{1e-6, 5e-6, 1e-5, 5e-5, 1e-4\}$. We used a single NVIDIA A100-40GB GPU for all our CLIP fine-tuning experiments. Our longest running experiments finished in just under 70 minutes.

### J.2  OTTER

For all Otter fine-tuning experiments, we froze the vision encoder, and only updated the perceiver resampler module, the cross-attention layers in the LLM-encoder, and the input/output embeddings. We fine-tuned the model on all data-type mixtures for 9 epochs with the AdamW optimizer, learning rate of $1e-5$, batch size of 128 and cosine-annealing learning-rate schedule, using accelerate-FSDP in mixed precision (`bfloat16`). We conducted our experiments on 6 NVIDIA A100-40GB GPUs and our longest running experiments finished in under 40 hours.

## K  IMAGE RIGHTS AND ATTRIBUTION

The images in Fig. 1 were sourced from various origins. We here attribute their original sources along with their corresponding licenses. We thank the original creators for availing these images for public use.

- A. All dog and eagle images are sourced from our *NaturalTypeIdent* dataset.
- B.
  - TYPOGRAPHIC image: https://www.goodhousekeeping.com/home-products/multi-purpose-cleaners/g579/best-multi-purpose-cleaners/
  - CARTOON image: https://www.kaggle.com/datasets/volkandl/cartoon-classification
    (license: https://creativecommons.org/publicdomain/zero/1.0/)
  - JPEG COMPRESS image: https://www.shutterbug.com/content/fix-ugly-jpegs-fast-photoshops-ai-filters-video
    (license: https://creativecommons.org/publicdomain/zero/1.0/)
- C.
  - SNOW image: https://copartautoblog.de/2019/12/19/driving-in-the-snow/
  - HIGH BRIGHTNESS image: https://www.defensivedriving.com/blog/driving-in-the-sun/

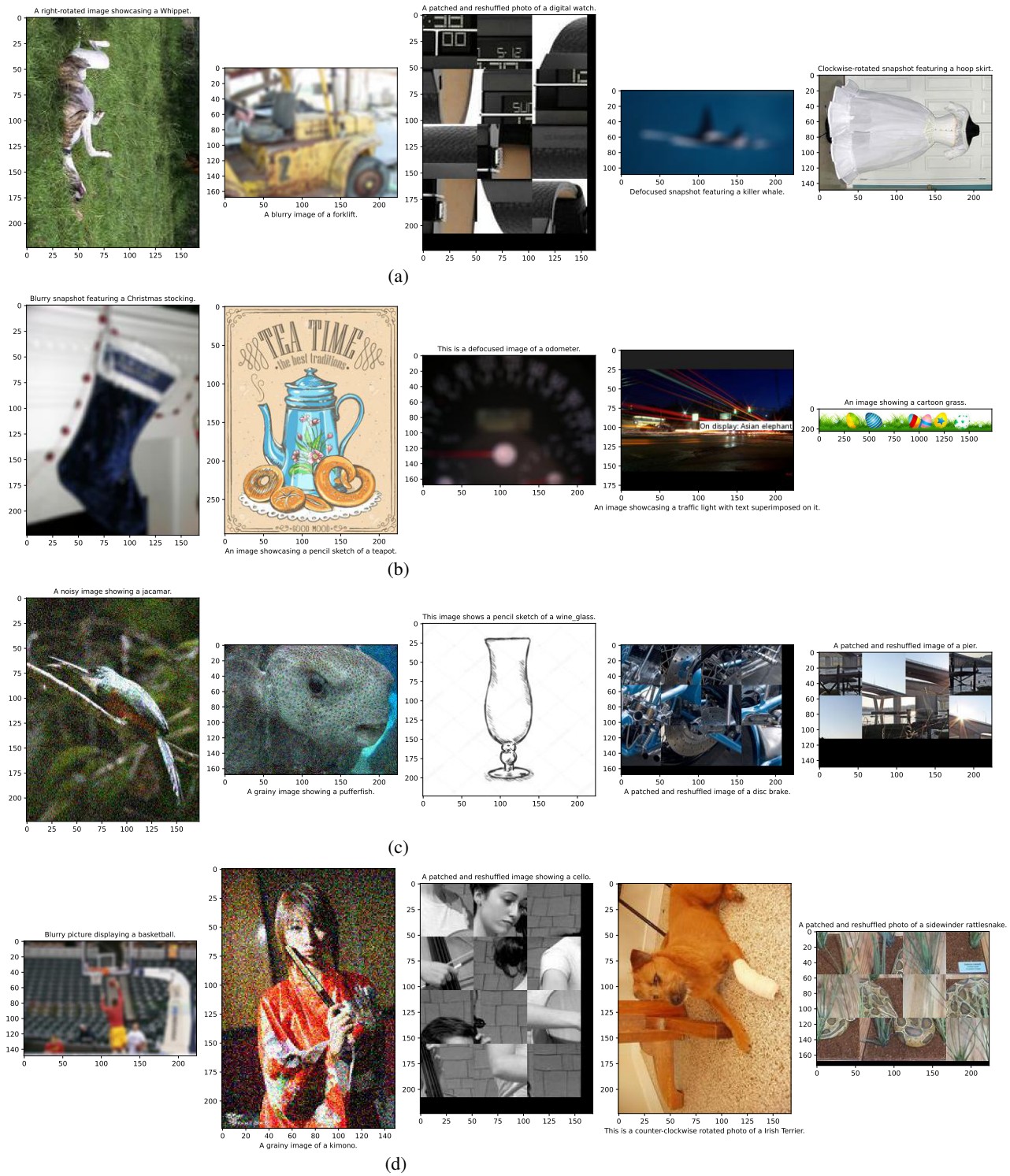

Figure 12: **Random samples from the TeDaTy dataset.**

- LEFT ROTATE image: https://jane-athome.com/long-narrow-living-room/

