# OpenReview forum: "Visual Data-Type Understanding does not emerge from scaling Vision-Language Models"
_ICLR.cc/2024/Conference — ICLR 2024 poster_

### Official Review · Reviewer_GzDz · 2023-10-18

**Soundness:** 1 poor
**Presentation:** 3 good
**Contribution:** 2 fair
**Rating:** 8
**Confidence:** 4

**Summary:**

This papers explores the task of visual data type identification to better understand abilities of vision-language models (VLMs). Data-type identification here refers to cases like distinguishing between a natural image and a left-rotation / blurred version of the natural image. The authors construct two datasets containing 27 different datatypes (under 4 categories of geometric, pixel, style, semantic) with one dataset synthetic and the other natural. The synthetic uses generative models and image manipulations for creating all its images. The natural uses web-crawled images. Considering VLMs under contrastive (C-VLMs) and generative (LMMs) categories, they evaluate 39 different VLMs on the task of distinguishing each of these different datatypes. Their results indicate weak performance of both categories, with (the generally larger, newer) LMMs inferior to C-VLMs. Interesting analysis is presented including analysis of feature spaces and fine-tuning these models with data-type based supervision (which gives good results).

**Strengths:**

1. Valuable dataset contribution
2. Interesting problem setup
3. Useful framework for creating more data-type datasets

4. Assuming correct, interesting and extensive analysis

The authors have performed extensive work evaluating a range of models and extracting multiple insights.

**Weaknesses:**

1. Method weak with respect to language (leading to possibly wrong analysis)

* A core component of experiments is language based classification (into one of 27 data-types). However, the authors simply use a short description for each data-type (as in appendix) on language side. This could simply mean that the model is not being prompted correctly (and the sensitivity of VLMs to prompting is well-known). If this is the case, the assertions of weaknesses (i.e. inability to distinguish data-types) is not generic. It is only for the selected set of prompts used in paper.
* For simple verification, I took 10 random images, applied the patch&reshuffle operation, and passed both original and augmented to the online LLava API (https://llava.hliu.cc - this is one of the models evaluated in paper / API uses newer ckpt). While it did not generate the expected term "patch&reshuffle", it generated outputs for the augmented images different to original, consistently involving terms like "collage, mosaic, collage art style" (words somewhat synonymous to augmentation) which indicate some awareness of the model to this patch&reshuffle operation. The original images were not described with these keywords. However, according to Figure 4 in paper, the best LMM (LLava is one of those evaluated) has 0 informedness about patch&reshuffle. For this case, either the metric of informedness or the evaluation protocol does not well-represent the actual abilities of evaluated models like LLava.


2. Missing LMM evaluation details

* "For a fair comparison, we evaluated LMMs by log-likelihood scoring" - please explain this in detail in the main text (or at least appendix) without directing reader to other papers (in fact the directed papers also re-direct to different papers) . This evaluation is crucial to understand all the reported analysis. Please explain it clearly.


While the paper possibly contains some flaws in their method / evaluation, I urge the authors to provide clear responses (especially in case I am mistaken). The dataset and evaluation framework contribution alone can be useful to the community. Also, the direction explored is highly meaningful and valuable.

---
*Post Rebuttal*
The authors provide extensive additional analysis to disprove concerns I raised, and also strengthen the paper detailing all information related to evaluation. Therefore, my final rating has been raised (5 to 8).

**Questions:**

1. KNN + Linear Probing
* For C-VLMs, can the analysis in Figure-5 be done with KNN evaluation and linear probing? This will give a clear understanding of how linearly separable the visual features are for each datatype. If they are separable, the idea that it is a misalignment with text modality will be clear, and this could be an interesting contribution. If they are not, maybe even SVM on features could be done as a final test to verify their lack of separation.
* While t-SNE embeddings are visually appealing, they can often be misleading in my personal experience.

2. VLM pre-training datasets
* While the text based search is good, it maybe better to use semantic similarity metrics (maybe word vectors, scores like METEOR) to create a more comprehensive vocabulary for each data type (e.g. PATCH AND RESHUFFLE can have more words like collage, mosaic). This could retrieve more images similar to the setup.

3. The reasoning for weakness
* Given how simple supervision helps a lot, the weaknesses could be attributed to a language-image mismatch in these models. Particularly, if the train set rarely contains these language terms. Can any simple experiments directly negate / verify this hypothesis?

---

> ### Author Response · Authors · 2023-11-12
> **Request for a clarification**
>
> Thank you for your insightful and constructive feedback. We greatly appreciate the effort you have invested in reviewing our paper, and we find your comments very helpful for improving our paper’s quality. Before we address all your concerns and questions, could you please further clarify your question on the language-image mismatch? Specifically, this question:
> > 3. **The reasoning for weakness**: Given how simple supervision helps a lot, the weaknesses could be attributed to a language-image mismatch in these models. Particularly, if the train set rarely contains these language terms. Can any simple experiments directly negate / verify this hypothesis?
>
> We are happy to provide additional experiments if possible, but are unsure what you meant.

---

> ### Comment · Reviewer_GzDz · 2023-11-12
>
> >The reasoning for weakness
>
> For visual-data type identification, the paper shows weak performance of VLMs (e.g. Figure 4), especially LMMs. Section 5 provides 2 reasons (CLIP image encoder and training datasets).  However, fine-tuning experiments with CLIP image encoder frozen (e.g. Table 1) provides strong performance for data-type identification. This indicates some feature separation by data-type (contrary to Sec 5 and Figure 5). Does this mean CLIP image encoder already has good task awareness (features are separated by data-type), but there is simply a mismatch between language features causing the observed weaknesses (for zero-shot probing)?
>
> My doubt here is that, maybe better language-based probing could solve the identified weaknesses. For example, if instead of simply using the manual prompt description (Appendix B.2) for evaluation, if we use more descriptive prompts for data types (it is shown how CLIP performance improves a lot with this [1]), this may overcome the VLM weaknesses identified. I would like this concern to be addressed.
>
> Please let me know if further clarification is needed.
>
> [1] Visual Classification via Description from Large Language Models (ICLR '23), https://cv.cs.columbia.edu/sachit/classviadescr

---

> > ### Comment · Reviewer_GzDz · 2023-11-21
> > **Cannot see author responses**
> >
> > I cannot see any author responses - are they hidden currently or simply no responses yet?

---

> ### Author Response · Authors · 2023-11-22
> **Response to reviewer GzDz**
>
> Thank you for thoroughly reviewing our paper, and providing your helpful comments and insightful questions. We highly appreciate your fast response and want to apologise for not replying earlier to you---we misunderstood the review process in that regard. We address all your comments and questions below.
>
> - **Regarding your first comment on prompting the LMMs and your third question on the reasoning for weakness.** We agree that reasonably prompting the LMMs is crucial for prompt-based model evaluation. Following your suggestions, we now have run additional experiments, verifying that the results reported in our paper did not emerge from a peculiarity in the prompts we used.
> 1. We created two additional sets of alternate prompts for the 27 data-types capturing language diversity. Inspired by VisDesc ([Menon and Vondrick ICLR, 2023](https://arxiv.org/abs/2210.07183)) and CuPL ([Pratt et al ICCV, 2023](https://arxiv.org/abs/2209.03320)), we prompted GPT-4 to suggest alternative prompts for each of the original data-type prompts and manually verified that they actually capture the correct data-type concept. For instance, for the PATCH_AND_RESHUFFLE data-type, in our initial experiments we probed VLMs with the prompt “This is a patched and reshuffled image of an animal”. For the two alternative prompt-sets we probe them with “This is a scrambled image of an animal.” and “This is a jumbled image of an animal.” (the full set of alternate prompts for all data-types is listed in Appendix B.3). Evaluating a representative subset of models on SyntheticTypeIdent revealed highest model performance for the prompts we originally chose, while the alternative prompts performed slightly worse (see new Appendix C and Figure 7 for results). Further, we were curious how the newer LLaVA 1.5 model you explored would perform on our task and included it into our systematic evaluation (both the 7B and 13B variants). In line with what you found in your experiments with the web-API, we found an improvement over LLaVA 1.0 (see Figure 7). At the same time, the absolute performance is comparable to the other LMMs tested.
> 2. To further extend this analysis, we followed your suggestion and used the GPT-3 prompting method introduced by the paper you cited (VisDesc ([Menon and Vondrick ICLR, 2023](https://arxiv.org/abs/2210.07183))), to generate a large set of data-type descriptors (5-7 descriptors per data-type). To probe C-VLMs with these descriptors, we followed the evaluation scheme by VisDesc: we averaged the corresponding text embeddings for each data-type and predicted the data-type whose embedding had the highest similarity with the test image. LMMs cannot be probed by comparing to averaged embeddings. Hence, we averaged the log-likelihood across descriptors for each data-type, akin to an evaluation strategy used for OpenFlamingo (see [Github](https://github.com/mlfoundations/open_flamingo/blob/655f693fbfa04cd6e9a987d960654624d48917cf/open_flamingo/eval/evaluate.py#L83C1-L87C2)). This much more sophisticated prompting method led to similar results as in our main analysis and the alternative prompt experiments (see new Appendix section C and Figure 8).
>
> In conclusion, while the exact prompting style showed some effect on model performance, overall prompt-based results on our data-type identification tasks (C-VLM outperforming LMMs, LMMs performing generally poor, only marginal scaling behaviour) robustly stayed the same across prompting strategies.
>
> - **Regarding LMM evaluation details.** We thank you for pointing out that this information was not clearly included in our paper. To compute the zero-shot performance of C-VLMs, we computed the matching score of each image $I$ with each of the 27 data-type text prompts $\{D_1, D_2, …, D_{27}\}$ and predicted the data type with the highest score, i.e. $\text{predicted data-type} = \arg \max_{i\in{1, …, 27}} I_{\text{enc}}(I) \cdot T_{\text{enc}}(D_i)$. Since LMMs are auto-regressively trained, evaluating them in the same way is impossible. For a fair comparison of LMMs to C-VLMs, we instead computed the LMMs log-likelihood of a prompt joining the image and the data-type text prompts. By default, we prompted LMMs to compute the log-likelihood for the query prompt $P_i =$ “<image> Q: Describe the image. A: $D_i$”, where $D_i$ is replaced by the specific data-type text prompt. Specifically, we tokenized the prompt $P_i$ and retrieved the model log-probabilities for each input token. Then, we summed the log-probabilities across tokens yielding the final log-probability for the particular data-type, $L_i(P_i(I, D_i))$. We predicted the data-type of an image $I$ by the highest log-probability across data-types, $\text{predicted data-type} = \arg \max_{i\in{1, …, 27}} L_i(P_i(I, D_i))$. Some LMMs were trained with a particular prompt pattern for instruction tuning, and for those, we used their specific prompt template $P$ (see Appendix B.5 for full list).

---

> ### Author Response · Authors · 2023-11-22
> **Response to reviewer GzDz continued**
>
> - **Regarding LMM evaluation details contd.** This log-likelihood evaluation strategy is common for evaluating LLMs on multiple choice questions ([Brown et al.](https://arxiv.org/abs/2005.14165), [Sanh et al.](https://arxiv.org/abs/2110.08207), https://blog.eleuther.ai/multiple-choice-normalization/) and LMMs on classification tasks ([InstructBLIP](https://github.com/salesforce/LAVIS/blob/7f00a0891b2890843f61c002a8e9532a40343648/lavis/models/blip2_models/blip2_t5_instruct.py#L503), [OpenFlamingo](https://github.com/mlfoundations/open_flamingo/blob/655f693fbfa04cd6e9a987d960654624d48917cf/open_flamingo/eval/models/open_flamingo.py#L155)). Further, as suggested by ([Brown et al.](https://arxiv.org/abs/2005.14165), https://blog.eleuther.ai/multiple-choice-normalization/) , we explored various length normalisation strategies to compute the log-likelihood scores and observed no significant changes to the results. Hence, we used the standard log-likelihood scoring procedure outlined above for all our results. We now have added new Appendix Section B.4 clearly describing this evaluation strategy and for concreteness showcase the exact code snippets for computing the log-probabilities of three of the models tested (See Appendix B.4 pages 25, 26 and 27). We think adding this information has significantly improved the clarity of our paper, thank you very much for this question.
>
> - **Regarding KNN and Linear Probing of CLIP visual features depicted in t-SNE Fig. 5.** We now have run KNN on the CLIP-RN50 image embeddings as features and animal-types / data-types as labels (shown in Fig. 5). While the KNN classifier performed well on identifying the animal class (0.96 mean informedness for k=1), performance significantly dropped in predicting the data-type (0.29 mean informedness for k=1; see new Appendix Table 5 for more values of k). Similarly, a linear probe (multinomial logistic regression with LBFGS optimizer akin to what was used for CLIP) showed high linear separability of animal classes (mean informedness=0.99), whereas data-types were less linearly separable (mean informedness=0.84). These values were obtained on the train set only, as for the question you raised, we want to study linear separability of the data points and are not interested in test set generalisation. The number of 1,350 data-points in our dataset is similar to the dimensionality of the CLIP-RN50’s feature space (1,024 dimensions), which might in part account for the higher performance of the logistic regression classifier. While we feel this is somewhat a limiting caveat of the linear probe analysis, we believe this analysis adds important information. Overall, the worse performance of these classifiers when applied on data-types compared to animal-types suggest CLIP is somewhat invariant to data-types. We now have added these results to Appendix F and new Table 5.
>
> - **Regarding your suggestion to use semantic similarity metrics to analyse the VLM pre-training datasets.** We agree that a purely text-based search might not fully capture a comprehensive data-type vocabulary from the pre-training dataset. Hence, we selected the [sentence-transformers/all-MiniLM-L6-v2](https://huggingface.co/sentence-transformers/all-MiniLM-L6-v2) text embedding model for performing semantic-similarity based search. This model is light-weight and optimised for effective sentence similarity matching in large corpora ([Reimers and Gurevych, EMNLP 2019](https://aclanthology.org/D19-1410.pdf)). With this model, we first encode the data-type text descriptions to get embeddings. We next encode the text captions in the LAION-2B-en pre-training dataset (due to compute limitations, we run this analysis only on 36 randomly selected parquet files out of 127, i.e. ~30% of the data). Leveraging these embeddings, we computed the cosine similarity between encoded data-type prompts and LAION text captions, keeping those samples that had a cosine similarity larger than 0.85. We chose 0.85 as our matching threshold by manually inspecting similarity scores on a curated set of sentences containing data-type descriptions---below 0.85 we noted that the concepts within two sentences no longer matched. The resulting rank correlation between abundancy scores and averaged model informedness evaluated on SyntheticTypeIdent were much higher compared to the simple text-search: r=0.798 for all models and r=0.776 for CLIP-based models only. These results support our previous results suggesting that lack of data-type rich samples in the pre-training data might lead to poor VLM performance for data-type identification. We have added these results into Appendix Section G.1.
>
> Thank you once again for all of your suggestions, we truly believe that they have helped significantly strengthen the results of our paper. We hope our added experimental results and analyses have addressed your questions and comments. We are happy to add further clarifications if needed.

---

### Official Review · Reviewer_mQDi · 2023-10-25

**Soundness:** 3 good
**Presentation:** 3 good
**Contribution:** 3 good
**Rating:** 8
**Confidence:** 3

**Summary:**

This research introduces the novel task of Visual Data-Type Identification, which involves identifying the visual data-type of images and holds practical value in data curation and autonomous vision systems. Two datasets were created, featuring animal images modified to represent 27 different visual data-types, and 39 VLMs were extensively evaluated. While VLMs perform well with certain stylistic data-types like cartoons and sketches, they face challenges with simpler data-types. Importantly, the study emphasizes that merely scaling up models is insufficient for improving performance, especially for the largest auto-regressively trained VLMs. By incorporating data-type information and pre-training analysis, the study achieved notable performance enhancements, setting the stage for advancing VLMs with visual data-type understanding.

**Strengths:**

1) The paper is clearly written and easy to follow.
2) The method introduces a novel task of Visual Data-Type Identification. This task involves recognizing the visual data-type of an image, such as whether an image has been altered, and how it has been changed. This concept is relatively unexplored in the field of vision-language models.
3) The researchers created two datasets containing animal images altered to represent 27 different visual data-types, spanning a wide range of categories. This diversity in the datasets allows for a more comprehensive evaluation of VLMs' performance. They conduct an extensive evaluation of 39 VLMs, covering a wide range of model sizes, from small to extremely large. This comprehensive evaluation provides insights into how different VLMs perform in the context of Visual Data-Type Identification.
4) The study identifies a limitation in existing VLMs. While these models excel at recognizing semantic content, they struggle to understand visual data-types, even when scaled up. This finding highlights the need for a more systematic approach to data-type understanding.
5) The method demonstrates a way to significantly enhance VLM performance by incorporating data-type information and pre-training analysis. This innovative approach improves the models' capability to understand visual data-types.

**Weaknesses:**

In page 6, the authors identify that LMMs consistently underperform C-VLMs, despite using
LLMs as text models, compared to the smaller text encoders in C-VLMs. The authors propose two potential factors for this difference, namely, "weak alignment" and the "discriminative-generative gap." However, it is suggested that these factors appear to be more like observations rather than fully explored reasons. It is recommended that further investigations are necessary to gain a deeper understanding of these performance differences.

**Questions:**

Please refer to the weakness part.

---

> ### Author Response · Authors · 2023-11-22
> **Response to reviewer mQDi**
>
> Thank you for your positive assessment of our paper. As you inferred, we formulate the hypothesis that weak alignment and the discriminative-generative gap could explain why LMMs underperform C-VLMs. Although beyond the scope of our study, related work provides some initial support for this hypothesis that could form the basis for future work ([Vapnik IEEE 1999](https://ieeexplore.ieee.org/document/788640), [Ng and Jordan, NeurIPS 2001](https://papers.nips.cc/paper_files/paper/2001/hash/7b7a53e239400a13bd6be6c91c4f6c4e-Abstract.html), [Saunders et al. arXiv 2022](https://arxiv.org/abs/2206.05802), [Bavishi et al. 2023](https://www.adept.ai/blog/fuyu-8b)). We now mention this more explicitly in the Results (section 4.2) and the Conclusion (section 7) of our paper.

---

### Official Review · Reviewer_aN8Q · 2023-11-05

**Soundness:** 3 good
**Presentation:** 3 good
**Contribution:** 3 good
**Rating:** 8
**Confidence:** 4

**Summary:**

The paper introduced a new task: visual data-type identification for vision foundation models. This task builds on earlier literature on robustness and domain adaptation of ImageNet models, but tailored for vision foundation models. This task has practical importance for data curation and data cleaning. The authors have conducted extensive experiments (detailed in the "Strengths" section below) and found that scaling model size results in minimal gain, and training with data-type information is a promising direction.

**Strengths:**

- Interesting taskification of data-type identification. I agree with the usefulness for downstream applications such as data curation and data cleaning.
- Introduced TypeIdent dataset spanning 27 data types across 4 categories (geometric, pixel, semantic, and style)
- Extensive experiments ranging from initial evaluation using 13 model families, error analysis using embeddings and looking into CLIP's pre-training dataset, in-context learning, and fine-tuning with newly created dataset TeDaTy, which incorporates data-type information into image-text pairs.
- Interesting findings such as: scaling model size results in minimal gain, in-context learning (using 7B models) doesn't improve the performance of data-type identification much.

**Weaknesses:**

-  The size of the model (i.e. 7B) used for in-context learning experiments might be too small to test the capability of in-context learning. In-context learning with larger models might work so I think it would be better if the authors could clarify this point.

**Questions:**

- For Section 4.1, when assessing the accuracy of LMMs, aren't there more than one correct answer for data type description? For example, "pencil sketch" could be "pencil drawing", "black and white drawing" etc and "Origami style" could be "low-poly style" etc. How did you deal with these?

---

> ### Author Response · Authors · 2023-11-22
> **Response to reviewer aN8Q**
>
> We thank the reviewer for the helpful comments and questions. We answer both below.
>
> - **Regarding the comment about prompt sensitivity.** Thank you for this great question! In our initial zero-shot results, we had used a default, fixed set of prompts for all our experiments. However, we now have run additional experiments to check for prompt sensitivity, verifying that the results reported in our paper did not emerge from a peculiarity in the prompts we used.
> 1. We created two additional sets of alternate prompts for the 27 data-types capturing language diversity. Inspired by VisDesc ([Menon and Vondrick, ICLR, 2023](https://arxiv.org/abs/2210.07183)) and CuPL ([Pratt et al, ICCV, 2023](https://arxiv.org/abs/2209.03320)), we prompted GPT-4 to suggest alternative prompts for each of the original data-type prompts and manually verified that they actually capture the correct data-type concept. For instance, for the PATCH_AND_RESHUFFLE data-type, in our initial experiments we probed VLMs with the prompt “This is a patched and reshuffled image of an animal”. For the two alternative prompt-sets we probe them with “This is a scrambled image of an animal.” and “This is a jumbled image of an animal.” (the full set of alternate prompts for all data-types is listed in Appendix B.3). Evaluating a representative subset of models on SyntheticTypeIdent revealed highest model performance for the prompts we originally chose, while the alternative prompts performed slightly worse (see new Appendix C and Figure 7 for results).
> 2. To further extend this analysis, we followed the GPT-3 prompting method introduced by VisDesc ([Menon and Vondrick, ICLR, 2023](https://arxiv.org/abs/2210.07183)), to generate a large set of data-type descriptors (5-7 descriptors per data-type). To probe C-VLMs with these descriptors, we followed the evaluation scheme by VisDesc: we averaged the corresponding text embeddings for each data-type and predicted the data-type whose embedding had the highest similarity with the test image. LMMs cannot be probed by comparing to averaged embeddings. Hence, we averaged the log-likelihood across descriptors for each data-type, akin to an evaluation strategy used for OpenFlamingo (see [Github](https://github.com/mlfoundations/open_flamingo/blob/655f693fbfa04cd6e9a987d960654624d48917cf/open_flamingo/eval/evaluate.py#L83C1-L87C2)). This much more sophisticated prompting method led to similar results as in our main analysis and the alternative prompt experiments (see new Appendix section C and Figure 8).
>
> In conclusion, while the exact prompting style showed some effect on model performance, overall prompt-based results on our data-type identification tasks (C-VLM outperforming LMMs, LMMs performing generally poor, only marginal scaling behaviour) robustly stayed the same across prompting strategies.
>
> - **Regarding in-context learning with larger models.** For Otter, which we previously investigated (Figure 6b), unfortunately no in-context learning model versions larger than the investigated 7B model exist. Thus, we repeated the same experiments as for Otter with the LLaVA models, where both 7B and 13B variants are available. Interestingly, both LLaVA versions underperformed Otter-LLaMA7B and Otter-MPT7B, and surprisingly, LLaVA-13B even underperformed its smaller LLaVA-7B version (new Appendix Section H.3 and Figure 11). This experiment further corroborates the claim of our paper that in-context learning seemingly does not provide benefits for improving the performance of LMMs like Otter and LLaVA on data-type identification. Thank you again for this great suggestion.

---

> > ### Comment · Reviewer_aN8Q · 2023-11-23
> > **Response**
> >
> > Thank you for your responses. My questions are resolved. I maintain my score.

---

### Author Response · Authors · 2023-11-22
**General response to all reviewers**

We thank all the reviewers for their insightful questions and comments. We have incorporated all suggestions, which we believe substantially strengthened our paper. We have made these main changes in the updated paper draft:
- Added two prompt variation experiments: (1) using two alternate sets of prompts, and (2) using descriptor enhanced prompts from GPT-3. See Appendix B.3 and C
- Added details about log-likelihood scoring for evaluating LMMs. See Appendix B.4
- Added K-NN and linear probing analysis on CLIP’s image space. See Appendix F
- Added semantic-similarity-based search correlation analysis instead of pure text-based search analysis. See Appendix G.1
- Added in-context learning scaling experiment with LLaVA 7B and 13B models. See Appendix H.3
- Revised text throughout the paper improving readability and added references to the new Appendix sections mentioned above.

We have provided detailed responses to each of your concerns individually below. Thank you once again for providing such great feedback.

---

### Meta-Review · Area_Chair_KFGX · 2023-12-11

**Metareview:**

In this paper the authors introduce a new task for multimodal foundational models. The goal of the task is to predict the type of data or the augmentation used to generate the data. The task speaks to central problems in vision such as robustness and domain adaptation but tailored to multimodal foundation models but is also immediately useful in terms of data curation. The reviewers commented positively on the quality of the task introduced by the authors, the extensiveness of the data-types and the comprehensiveness of the experiments across vision language models (VLMs). The reviewers also raised concerns requesting additional experiments and analysis of the language component of the VLMs, as well as the overall size of the model. All of these concerns were addressed in the rebuttals. Given that the paper focuses on an impactful question and provides new methods for understanding multimodal foundational models, this work will be accepted at this conference.

**Justification For Why Not Higher Score:**

n/a

**Justification For Why Not Lower Score:**

Important new evaluation regime for large models. Good experiments.

---

### Decision · Program_Chairs · 2024-01-16

Accept (poster)